# EMERGENT TOOL USE FROM MULTI-AGENT AUTOCURRICULA

**Bowen Baker**\*
OpenAI
bowen@openai.com

**Ingmar Kanitscheider**\*
OpenAI
ingmar@openai.com

**Todor Markov**\*
OpenAI
todor@openai.com

**Yi Wu**\*
OpenAI
jxwuyi@openai.com

**Glenn Powell**\*
OpenAI
glenn@openai.com

**Bob McGrew**\*
OpenAI
bmcgrew@openai.com

**Igor Mordatch**\*†
Google Brain
imordatch@google.com

## ABSTRACT

Through multi-agent competition, the simple objective of *hide-and-seek*, and standard reinforcement learning algorithms at scale, we find that agents create a self-supervised autocurriculum inducing multiple distinct rounds of emergent strategy, many of which require sophisticated tool use and coordination. We find clear evidence of six emergent phases in agent strategy in our environment, each of which creates a new pressure for the opposing team to adapt; for instance, agents learn to build multi-object shelters using moveable boxes which in turn leads to agents discovering that they can overcome obstacles using ramps. We further provide evidence that multi-agent competition may scale better with increasing environment complexity and leads to behavior that centers around far more human-relevant skills than other self-supervised reinforcement learning methods such as intrinsic motivation. Finally, we propose transfer and fine-tuning as a way to quantitatively evaluate targeted capabilities, and we compare hide-and-seek agents to both intrinsic motivation and random initialization baselines in a suite of domain-specific intelligence tests.

## 1 INTRODUCTION

Creating intelligent artificial agents that can solve a wide variety of complex human-relevant tasks has been a long-standing challenge in the artificial intelligence community. Of particular relevance to humans will be agents that can sense and interact with objects in a physical world. One approach to creating these agents is to explicitly specify desired tasks and train a reinforcement learning (RL) agent to solve them. On this front, there has been much recent progress in solving physically grounded tasks, e.g. dexterous in-hand manipulation (Rajeswaran et al., 2017; Andrychowicz et al., 2018) or locomotion of complex bodies (Schulman et al., 2015; Heess et al., 2017). However, specifying reward functions or collecting demonstrations in order to supervise these tasks can be

---

\*This was a large project and many people made significant contributions. Bowen, Bob, and Igor conceived the project and provided guidance through all stages of the work. Bowen created the initial environment, infrastructure and models, and obtained the first results of sequential skill progression. Ingmar obtained the first results of tool use, contributed to environment variants, created domain-specific statistics, and with Bowen created the final environment. Todor created the manipulation tasks in the transfer suite, helped Yi with the RND baseline, and prepared code for open-sourcing. Yi created the navigation tasks in the transfer suite, intrinsic motivation comparisons, and contributed to environment variants. Glenn contributed to designing the final environment and created final renderings and project video. Igor provided research supervision and team leadership.

†Work performed while at OpenAI

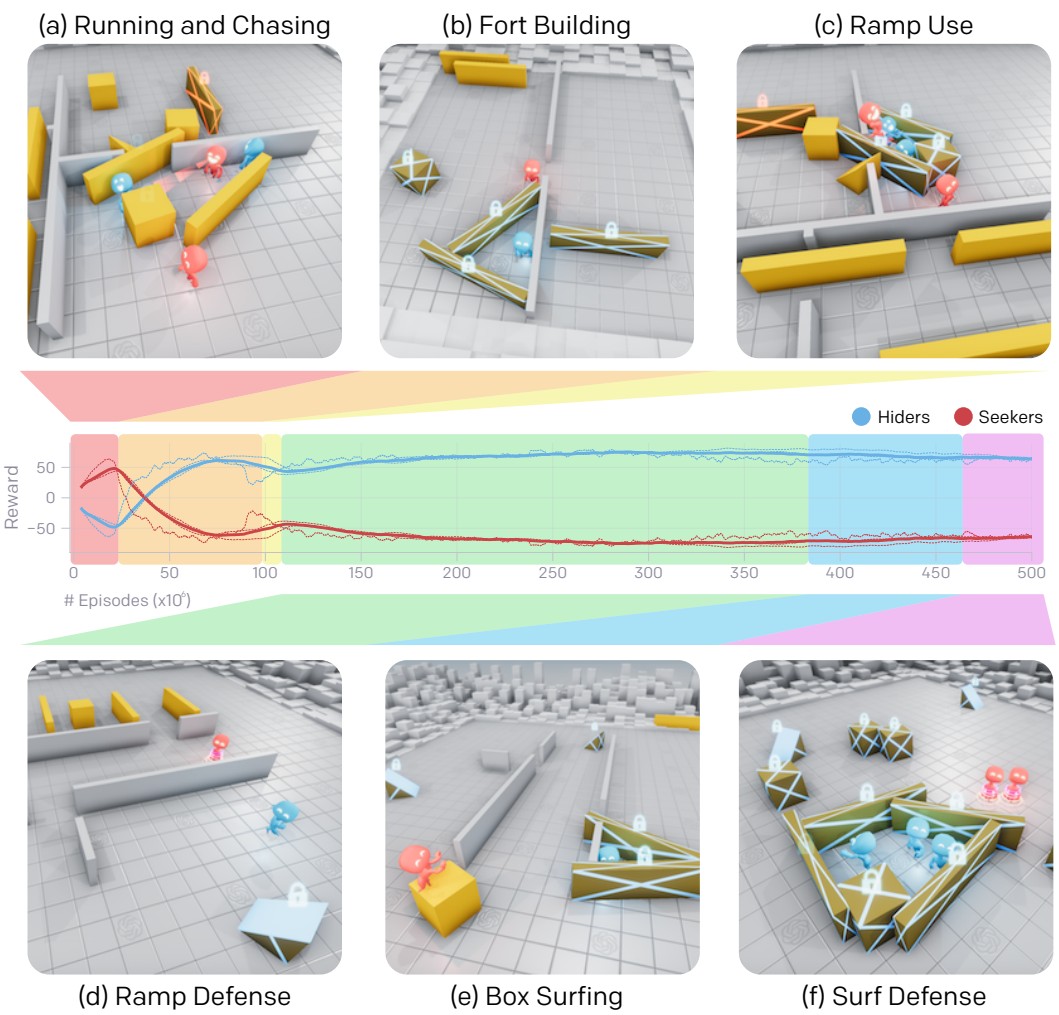

Figure 1: Emergent Skill Progression From Multi-Agent Autocurricula. Through the reward signal of hide-and-seek (shown on the y-axis), agents go through 6 distinct stages of emergence. (a) Seekers (red) learn to chase hiders, and hiders learn to crudely run away. (b) Hiders (blue) learn basic tool use, using boxes and sometimes existing walls to construct forts. (c) Seekers learn to use ramps to jump into the hiders' shelter. (d) Hiders quickly learn to move ramps to the edge of the play area, far from where they will build their fort, and lock them in place. (e) Seekers learn that they can jump from locked ramps to unlocked boxes and then *surf* the box to the hiders' shelter, which is possible because the environment allows agents to move together with the box regardless of whether they are on the ground or not. (f) Hiders learn to lock all the unused boxes before constructing their fort. We plot the mean over 3 independent training runs with each individual seed shown with a dotted line. Please see openai.com/blog/emergent-tool-use for example videos.

time consuming and costly. Furthermore, the learned skills in these single-agent RL settings are inherently bounded by the task description; once the agent has learned to solve the task, there is little room to improve.

Due to the high likelihood that direct supervision will not scale to unboundedly complex tasks, many have worked on unsupervised exploration and skill acquisition methods such as intrinsic motivation. However, current undirected exploration methods scale poorly with environment complexity and are drastically different from the way organisms evolve on Earth. The vast amount of complexity and diversity on Earth evolved due to co-evolution and competition between organisms, directed by natural selection (Dawkins & Krebs, 1979). When a new successful strategy or mutation emerges, it changes the implicit task distribution neighboring agents need to solve and creates a new pressure

for adaptation. These evolutionary arms races create implicit *autocurricula* (Leibo et al., 2019a) whereby competing agents continually create new tasks for each other. There has been much success in leveraging multi-agent autocurricula to solve multi-player games, both in classic discrete games such as Backgammon (Tesauro, 1995) and Go (Silver et al., 2017), as well as in continuous real-time domains such as Dota (OpenAI, 2018) and Starcraft (Vinyals et al., 2019). Despite the impressive emergent complexity in these environments, the learned behavior is quite abstract and disembodied from the physical world. Our work sees itself in the tradition of previous studies that showcase emergent complexity in simple physically grounded environments (Sims, 1994a; Bansal et al., 2018; Jaderberg et al., 2019; Liu et al., 2019); the success in these settings inspires confidence that inducing autocurricula in physically grounded and open-ended environments could eventually enable agents to acquire an unbounded number of human-relevant skills.

We introduce a new mixed competitive and cooperative physics-based environment in which agents compete in a simple game of hide-and-seek. Through only a visibility-based reward function and competition, agents learn many emergent skills and strategies including collaborative tool use, where agents intentionally change their environment to suit their needs. For example, hiders learn to create shelter from the seekers by barricading doors or constructing multi-object forts, and as a counter strategy seekers learn to use ramps to jump into hiders' shelter. Moreover, we observe signs of dynamic and growing complexity resulting from multi-agent competition and standard reinforcement learning algorithms; we find that agents go through as many as six distinct adaptations of strategy and counter-strategy, which are depicted in Figure 1. We further present evidence that multi-agent co-adaptation may scale better with environment complexity and qualitatively centers around more human-interpretable behavior than intrinsically motivated agents.

However, as environments increase in scale and multi-agent autocurricula become more open-ended, evaluating progress by qualitative observation will become intractable. We therefore propose a suite of targeted intelligence tests to measure capabilities in our environment that we believe our agents may eventually learn, e.g. object permanence (Baillargeon & Carey, 2012), navigation, and construction. We find that for a number of the tests, agents pretrained in hide-and-seek learn faster or achieve higher final performance than agents trained from scratch or pretrained with intrinsic motivation; however, we find that the performance differences are not drastic, indicating that much of the skill and feature representations learned in hide-and-seek are entangled and hard to fine-tune.

The main contributions of this work are: 1) clear evidence that multi-agent self-play can lead to emergent autocurricula with many distinct and compounding phase shifts in agent strategy, 2) evidence that when induced in a physically grounded environment, multi-agent autocurricula can lead to human-relevant skills such as tool use, 3) a proposal to use transfer as a framework for evaluating agents in open-ended environments as well as a suite of targeted intelligence tests for our domain, and 4) open-sourced environments and code[1] for environment construction to encourage further research in physically grounded multi-agent autocurricula.

## 2 RELATED WORK

There is a long history of using self-play in multi-agent settings. Early work explored self-play using genetic algorithms (Paredis, 1995; Pollack et al., 1997; Rosin & Belew, 1995; Stanley & Miikkulainen, 2004). Sims (1994a) and Sims (1994b) studied the emergent complexity in morphology and behavior of creatures that coevolved in a simulated 3D world. Open-ended evolution was further explored in the environments Polyworld (Yaeger, 1994) and Geb (Channon et al., 1998), where agents compete and mate in a 2D world, and in Tierra (Ray, 1992) and Avida (Ofria & Wilke, 2004), where computer programs compete for computational resources. More recent work attempted to formulate necessary preconditions for open-ended evolution (Taylor, 2015; Soros & Stanley, 2014). Co-adaptation between agents and environments can also give rise to emergent complexity (Florensa et al., 2017; Sukhbaatar et al., 2018; Wang et al., 2019). In the context of multi-agent RL, Tesauro (1995), Silver et al. (2016), OpenAI (2018), Jaderberg et al. (2019) and Vinyals et al. (2019) used self-play with deep RL techniques to achieve super-human performance in Backgammon, Go, Dota, Capture-the-Flag and Starcraft, respectively. Bansal et al. (2018) trained agents in a simulated 3D physics environment to compete in various games such as sumo wrestling and soccer goal shooting. In Liu et al. (2019), agents learn to manipulate a soccer ball in a 3D soccer environment and discover

---

[1]Code can be found at `github.com/openai/multi-agent-emergence-environments`.

emergent behaviors such as ball passing and interception. In addition, communication has also been shown to emerge from multi-agent RL (Sukhbaatar et al., 2016; Foerster et al., 2016; Lowe et al., 2017; Mordatch & Abbeel, 2018).

Intrinsic motivation methods have been widely studied in the literature (Chentanez et al., 2005; Singh et al., 2010). One example is count-based exploration, where agents are incentivized to reach infrequently visited states by maintaining state visitation counts (Strehl & Littman, 2008; Bellemare et al., 2016; Tang et al., 2017) or density estimators (Ostrovski et al., 2017; Burda et al., 2019b). Another paradigm are transition-based methods, in which agents are rewarded for high prediction error in a learned forward or inverse dynamics model (Schmidhuber, 1991; Stadie et al., 2015; Mohamed & Rezende, 2015; Houthooft et al., 2016; Achiam & Sastry, 2017; Pathak et al., 2017; Burda et al., 2019a; Haber et al., 2018). Jaques et al. (2019) consider multi-agent scenarios and adopt causal influence as a motivation for coordination. In our work, we utilize intrinsic motivation methods as an alternative exploration baseline to multi-agent autocurricula. Similar comparisons have also been made in Haber et al. (2018) and Leibo et al. (2019b).

Tool use is a hallmark of human and animal intelligence (Hunt, 1996; Shumaker et al., 2011); however, learning tool use in RL settings can be a hard exploration problem when rewards are unaligned. For example, in Forestier et al. (2017); Xie et al. (2019) a real-world robot learns to solve various tasks requiring tools. In Bapst et al. (2019), an agent solves construction tasks in a 2-D environment using both model-based and model-free methods. Allen et al. (2019) uses a combination of human-designed priors and model-based policy optimization to solve a collection of physics-based puzzles requiring tool use. However, in each of these works, agents were explicitly incentivized to interact with and use tools, whereas in our environment agents implicitly create this incentive through multi-agent competition.

## 3 HIDE AND SEEK

Agents are tasked with competing in a two-team hide-and-seek game in a physics-based environment. The *hiders* are tasked with avoiding line of sight from the *seekers*, and the seekers are tasked with keeping vision of the hiders. There are objects scattered throughout the environment that the agents can grab and also lock in place. There are also randomly generated immovable rooms and walls that the agents must learn to navigate. Before the game of hide-and-seek begins, the hiders are given a *preparation phase* where the seekers are immobilized, giving the hiders a chance to run away or change their environment.

There are no explicit incentives for agents to interact with objects in the environment; the only supervision given is through the hide-and-seek objective. Agents are given a *team based* reward; hiders are given a reward of 1 if all hiders are hidden and -1 if any hider is seen by a seeker. Seekers are given the opposite reward, -1 if all hiders are hidden and +1 otherwise. To confine agent behavior to a reasonable space, agents are penalized with a reward of -10 if they go too far outside of the play area (outside an 18 meter square). An episode lasts 240 timesteps, the first 40% of which are the preparation phase where all agents are given zero reward.

We simulate the environment in the MUJOCO physics engine (Todorov et al., 2012). The world is populated with 1 to 3 hiders, 1 to 3 seekers, 3 to 9 movable boxes of which at least 3 are elongated, 2 movable ramps, and randomly generated static walls and rooms. We also experiment with a simpler, less randomized environment described in Appendix A.3. Agents observe the position, velocity, and size (in the case of the randomly shaped boxes) of objects and other agents. If entities are not in line-of-sight of the agent or not in a 135 degree cone in front of the agent, then they are masked out in the policy. Agents also have 30 range sensors arrayed evenly around them, similar to a lidar. In addition, each agent observes its own team and other agents' teams as well as how much time is left in the preparation phase.

Agents are simulated as spherical objects and have 3 action types that can be chosen simultaneously at each time step. They may *move* by setting a discretized force along their $x$ and $y$ axis and torque around their $z$-axis. They have a single binary action to *grab* objects, which binds the agent to the closest object while the action is enabled. Agents may also *lock* objects in place with a single binary action. Objects may be unlocked only by agents on the team of the agent who originally locked the object. Agents may only grab or lock objects that are in front of them and within a small radius.

## 4 POLICY OPTIMIZATION

Agents are trained using self-play, which acts as a natural curriculum as agents always play opponents of an appropriate level.

Agent policies are composed of two separate networks with different parameters – a policy network which produces an action distribution and a critic network which predicts the discounted future returns. Policies are optimized using Proximal Policy Optimization (PPO) (Schulman et al., 2017) and Generalized Advantage Estimation (GAE) (Schulman et al., 2015), and training is performed using *rapid* (OpenAI, 2018), a large-scale distributed RL framework. We utilize decentralized execution and centralized training. At execution time, each agent acts given only its own observations and memory state. At optimization time, we use a centralized omniscient value function for each agent, which has access to the full environment state without any information masked due to visibility, similar to Pinto et al. (2017); Lowe et al. (2017); Foerster et al. (2018).

In all reported experiments, agents share the same policy parameters but act and observe independently; however, we found using separate policy parameters per agent also achieved all six stages of emergence but at reduced sample efficiency.

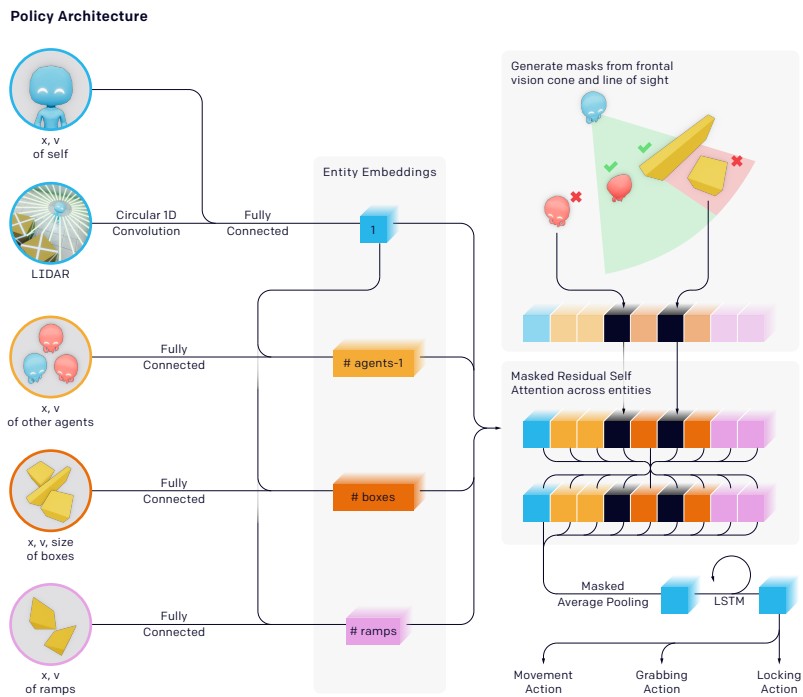

Figure 2: Agent Policy Architecture. All entities are embedded with fully connected layers with shared weights across entity types, e.g. all box entities are encoded with the same function. The policy is ego-centric so there is only one embedding of "self" and ($\#$agents $-1$) embeddings of other agents. Embeddings are then concatenated and processed with masked residual self-attention and pooled into a fixed sized vector (all of which admits a variable number of entities). $x$ and $v$ stand for state (position and orientation) and velocity.

We utilize entity-centric observations (Džeroski et al., 2001; Diuk et al., 2008) and use attention mechanisms to capture object-level information (Duan et al., 2017; Zambaldi et al., 2018). As shown in Figure 2 we use a self-attention (Vaswani et al., 2017) based policy architecture over entities, which is permutation invariant and generalizes to varying number of entities. More details can be found in Appendix B.

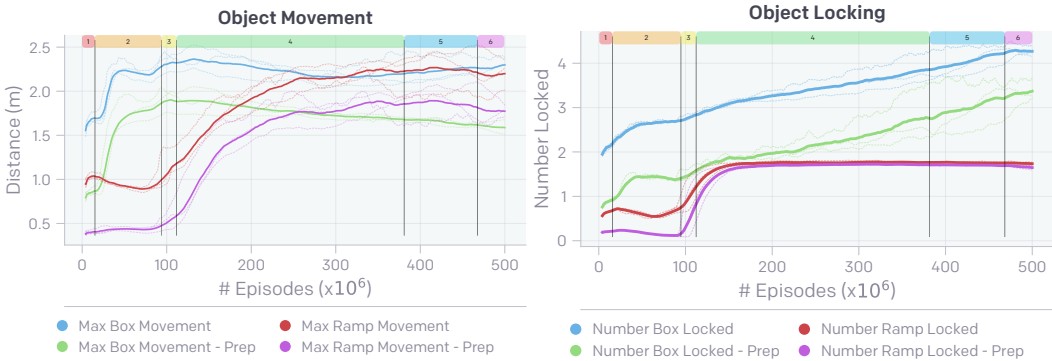

Figure 3: Environment specific statistics used to track stages of emergence in hide-and-seek. We plot the mean across 3 seeds with each individual seed shown in a dotted line, and we overlay the 6 emergent phases of strategy: (1) Running and Chasing, (2) Fort Building, (3) Ramp Use, (4) Ramp Defense, (5) Box Surfing, (6) Surf Defense. We track the maximum movement of any box or ramp during the game as well as during the preparation phase (denoted with "Prep"). We similarly track how many objects of each type were locked at the end of the episode and preparation phase. As agents train, their interaction with the tools in their environment changes. For instance, as the agents learn to build forts they move boxes and lock boxes much more during the preparation phase.

## 5  AUTO-CURRICULA AND EMERGENT BEHAVIOR

As agents train against each other in hide-and-seek, as many as six distinct strategies emerge, each of which creates a previously non-existing pressure for agents to progress to the next stage. Note that there are no direct incentives for agents to interact with objects or to explore, but rather the emergent strategies are solely a result of the autocurriculum induced by multi-agent competition. Figure 1 shows the progression of emergent strategies agents learn in our environment (see Appendix A.1 for trajectory traces of each strategy[2]).

Initially, hiders and seekers learn to crudely run away and chase. After approximately 25 million episodes of hide-and-seek, the hiders learn to use the tools at their disposal and intentionally modify their environment. They begin to construct secure shelters in which to hide by moving many boxes together or against walls and locking them in place. After another 75 million episodes, the seekers also learn rudimentary tool use; they learn to move and use ramps to jump over obstacles, allowing them to enter the hiders' shelter. 10 million episodes later, the hiders learn to defend against this strategy; the hiders learn to bring the ramps to the edge of the play area and lock them in place, seemingly removing the only tool the seekers have at their disposal.

We originally believed defending against ramp use would be the last stage of emergence in this environment; however, we were surprised to find that yet two more qualitatively new strategies emerged. After 380 million total episodes of training, the seekers learn to bring a box to the edge of the play area where the hiders have locked the ramps. The seekers then use the ramp to move on top of the box and *surf* it to the hiders' shelter. This is possible because the agents' movement action allows them to apply a force on themselves regardless of whether they are on the ground or not; if they do this while grabbing the box under them, the box will move with while they are on top of it. In response, the hiders learn to lock all of the boxes in place before building their shelter.[3]

In all stages of strategy agents must learn to coordinate within their team. Similar to Liu et al. (2019), we use team-based rewards such that agents are required to collaborate in order to succeed; however, in our work we require neither population-based training (Jaderberg et al., 2017) or evolved dense rewards (Jaderberg et al., 2019). Notably, hiders learn efficient division of labor; for instance, when constructing shelter they often separately bring their own box to the construction area. Furthermore,

---

[2]See openai.com/blog/emergent-tool-use for sample videos.

[3]Note that the discovery of a new skill does not necessarily correlate to the reward of a team changing. For example, the hider reward still decreases even after the discovery of surf defense, which is likely because teams with one or two hiders often do not have enough time to lock all of the boxes in play.

as hiders attempt to defend against box surfing by locking boxes in place during the preparation phase, we find that 2 and 3 hider teams are able to lock 25% and 36% more boxes, respectively, than a single hider team, indicating that larger teams are able to divide labor and accomplish an objective a smaller team could not.

Many stages of emergent strategy can be mapped to behavioral shifts in the way agents interact with the tools in their environment, similar to Leibo et al. (2017); Perolat et al. (2017). We therefore track basic statistics about the agents' interaction with objects during training, shown in Figure 3. For instance, as the hiders learn to build forts, they move and lock boxes much more during the preparation phase. Similarly, as the seekers learn to move and use ramps, the ramp movement in the main phase of the game increases, and as they learn to "box surf" there is a slight increase in the box movement during the main phase of the game. Finally, as the hiders learn to defend against this strategy by locking all boxes in place, the number of locked boxes in the preparation phase increases.

We found that scale plays a critical role in enabling progression through the emergent autocurricula in hide-and-seek. The default model, which uses a batch size of 64,000 and 1.6 million parameters, requires 132.3 million episodes (31.7 billion frames) over 34 hours of training to reach stage 4 of the skill progression, i.e. ramp defense. In Figure 4 we show the effect of varying the batch size in our agents ability to reach stage 4. We find that larger batch sizes lead to much quicker training time by virtue of reducing the number of required optimization steps, while only marginally affecting sample efficiency down to a batch size of 32,000; however, we found that experiments with batch sizes of 16,000 and 8,000 never converged.

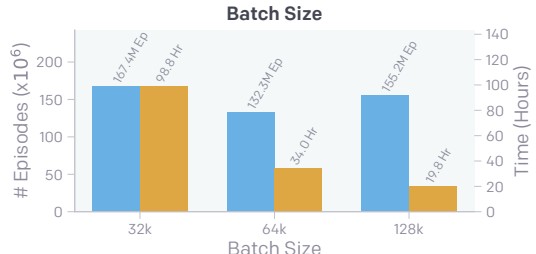

Figure 4: Effect of Scale on Emergent Autocurricula. Number of episodes (blue) and wall clock time (orange) required to achieve stage 4 (ramp defense) of the emergent skill progression presented in Figure 1. Batch size denotes number of chunks, each of which consists of 10 contiguous transitions (the truncation length for backpropagation through time).

We find the emergent autocurriculum to be fairly robust as long as we randomize the environment during training. If randomization is reduced, we find that fewer stages of the skill progression emerges, and at times less sophisticated strategies emerge instead (e.g. hiders can learn to run away and use boxes as moveable shields.); see Appendix A.2 for more details. In addition, we find that design choices such as the minimum number of elongated boxes or giving each agent their own locking mechanism instead of a team based locking mechanism can drastically increase the sample complexity. We also experimented with adding additional objects and objectives to our hide-and-seek environment as well as with several game variants instead of hide-and-seek (see Appendix A.6). We find that these alternative environments also lead to emergent tool use, providing further evidence that multi-agent interaction is a promising path towards self-supervised skill acquisition.

## 6 EVALUATION

In the previous section we presented evidence that hide-and-seek induces a multi-agent autocurriculum such that agents continuously learn new skills and strategies. As is the case with many unsupervised reinforcement learning methods, the objective being optimized does not directly incentivize the learned behavior, making evaluation of those behaviors nontrivial. Tracking reward is an insufficient evaluation metric in multi-agent settings, as it can be ambiguous in indicating whether agents are improving evenly or have stagnated. Metrics like ELO (Elo, 1978) or Trueskill (Herbrich et al., 2007) can more reliably measure whether performance is improving relative to previous policy versions or other policies in a population; however, these metrics still do not give insight into whether improved performance stems from new adaptations or improving previously learned skills. Finally, using environment specific statistics such as object movement (see Figure 3) can also be ambiguous, e.g. the choice to track absolute movement does not illuminate which direction agents moved, and designing sufficient metrics will become difficult and costly as environments scale.

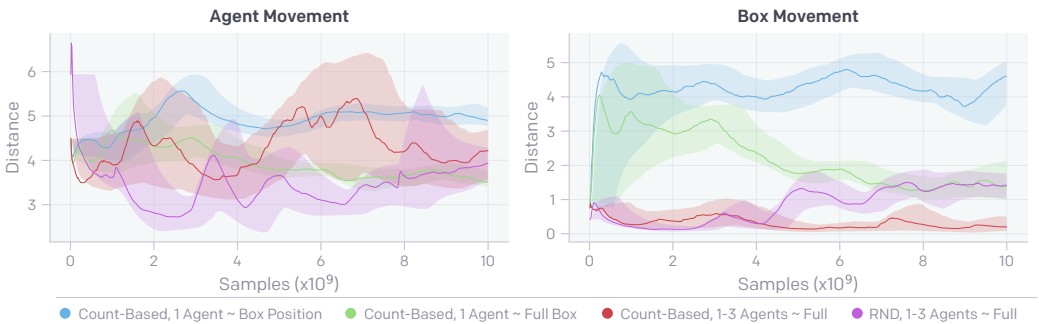

Figure 5: Behavioral Statistics from Count-Based Exploration Variants and Random Network Distillation (RND) Across 3 Seeds. We compare net box movement and maximum agent movement between state representations for count-based exploration: Single agent, 2-D box location (blue); Single agent, box location, rotation and velocity (green); 1-3 agents, full observation space (red). Also shown is RND for 1-3 agents with full observation space (purple). We train all agents to convergence as measured by their behavioral statistics.

In Section 6.1, we first qualitatively compare the behaviors learned in hide-and-seek to those learned from intrinsic motivation, a common paradigm for unsupervised exploration and skill acquisition. In Section 6.2, we then propose a suite of domain-specific intelligence tests to quantitatively measure and compare agent capabilities.

## 6.1 COMPARISON TO INTRINSIC MOTIVATION

Intrinsic motivation has become a popular paradigm for incentivizing unsupervised exploration and skill discovery, and there has been recent success in using intrinsic motivation to make progress in sparsely rewarded settings (Bellemare et al., 2016; Burda et al., 2019b). Because intrinsically motivated agents are incentivized to explore uniformly, it is conceivable that they may not have meaningful interactions with the environment (as with the "noisy-TV" problem (Burda et al., 2019a)). As a proxy for comparing meaningful interaction in the environment, we measure agent and object movement over the course of an episode.

We first compare behaviors learned in hide-and-seek to a count-based exploration baseline (Strehl & Littman, 2008) with an object invariant state representation, which is computed in a similar way as in the policy architecture in Figure 2. Count-based objectives are the simplest form of state density based incentives, where one explicitly keeps track of state visitation counts and rewards agents for reaching infrequently visited states (details can be found in Appendix D). In contrast to the original hide-and-seek environment where the initial locations of agents and objects are randomized, we restrict the initial locations to a quarter of the game area to ensure that the intrinsically motivated agents receive additional rewards for exploring.

We find that count-based exploration leads to the largest agent and box movement if the state representation only contains the 2-D location of boxes: the agent consistently interacts with objects and learns to navigate. Yet, when using progressively higher-dimensional state representations, such as box location, rotation and velocity or 1-3 agents with full observation space, agent movement and, in particular, box movement decrease substantially. This is a severe limitation because it indicates that, when faced with highly complex environments, count-based exploration techniques require identifying by hand the "interesting" dimensions in state space that are relevant for the behaviors one would like the agents to discover. Conversely, multi-agent self-play does not need this degree of supervision. We also train agents with random network distillation (RND) (Burda et al., 2019b), an intrinsic motivation method designed for high dimensional observation spaces, and find it to perform slightly better than count-based exploration in the full state setting.

## 6.2 TRANSFER AND FINE-TUNING AS EVALUATION

We propose to use transfer to a suite of domain-specific tasks in order to asses agent capabilities. To this end, we have created 5 benchmark intelligence tests that include both supervised and reinforce-

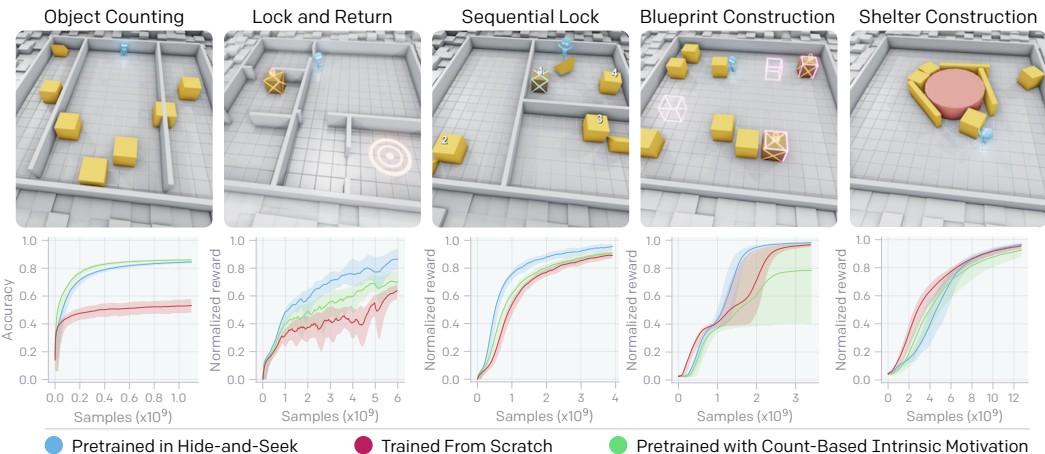

Figure 6: Fine-tuning Results. We plot the mean normalized performance and 90% confidence interval across 3 seeds smoothed with an exponential moving average, except for Blueprint Construction where we plot over 6 seeds due to higher training variance.

ment learning tasks. The tests use the same action space, observation space, and types of objects as in the hide-and-seek environment. We examine whether pretraining agents in our multi-agent environment and then fine-tuning them on the evaluation suite leads to faster convergence or improved overall performance compared to training from scratch or pretraining with count-based intrinsic motivation. We find that on 3 out of 5 tasks, agents pretrained in the hide-and-seek environment learn faster and achieve a higher final reward than both baselines.

We categorize the 5 intelligence tests into 2 domains: cognition and memory tasks, and manipulation tasks. We briefly describe the tasks here; for the full task descriptions, see Appendix C. For all tasks, we reinitialize the parameters of the final dense layer and layernorm for both the policy and value networks.

**Cognition and memory tasks:**

In the *Object Counting* supervised task, we aim to measure whether the agents have a sense of object permanence; the agent is pinned to a location and watches as 6 boxes each randomly move to the right or left where they eventually become obscured by a wall. It is then asked to predict how many boxes have gone to each side for many timesteps after all boxes have disappeared. The agent's policy parameters are frozen and we initialize a classification head off of the LSTM hidden state. In the baseline, the policy network has frozen random parameters and only the classification head off of the LSTM hidden state is trained.

In *Lock and Return* we aim to measure whether the agent can remember its original position while performing a new task. The agent must navigate an environment with 6 random rooms and 1 box, lock the box, and return to its starting position.

In *Sequential Lock* there are 4 boxes randomly placed in 3 random rooms without doors but with a ramp in each room. The agent needs to lock all the boxes in a particular order — a box is only lockable when it is locked in the correct order — which is unobserved by the agent. The agent must discover the order, remember the position and status of visited boxes, and use ramps to navigate between rooms in order to finish the task efficiently.

**Manipulation tasks:** With these tasks we aim to measure whether the agents have any latent skill or representation useful for manipulating objects.

In the *Construction From Blueprint* task, there are 8 cubic boxes in an open room and between 1 and 4 target sites. The agent is tasked with placing a box on each target site.

In the *Shelter Construction* task there are 3 elongated boxes, 5 cubic boxes, and one static cylinder. The agent is tasked with building a shelter around the cylinder.

**Results:** In Figure 6 we show the performance on the suite of tasks for the hide-and-seek, count-based, and trained from scratch policies across 3 seeds. The hide-and-seek pretrained policy performs slightly better than both the count-based and the randomly initialized baselines in *Lock and Return*, *Sequential Lock* and *Construction from Blueprint*; however, it performs slightly worse than the count-based baseline on *Object Counting*, and it achieves the same final reward but learns slightly slower than the randomly initialized baseline on *Shelter Construction*.

We believe the cause for the mixed transfer results is rooted in agents learning skill representations that are entangled and difficult to fine-tune. We conjecture that tasks where hide-and-seek pretraining outperforms the baseline are due to reuse of learned feature representations, whereas better-than-baseline transfer on the remaining tasks would require reuse of learned skills, which is much more difficult. This evaluation metric highlights the need for developing techniques to reuse skills effectively from a policy trained in one environment to another. In addition, as future environments become more diverse and agents must use skills in more contexts, we may see more generalizable skill representations and more significant signal in this evaluation approach.

In Appendix A.5 we further evaluate policies sampled during each phase of emergent strategy on the suite of targeted intelligence tasks, by which we can gain intuition as to whether the capabilities we measure improve with training, are transient and accentuated during specific phases, or generally uncorrelated to progressing through the autocurriculum. Notably, we find the agent's memory improves through training as indicated by performance in the navigation tasks; however, performance in the manipulation tasks is uncorrelated, and performance in object counting changes seems transient with respect to source hide-and-seek performance.

## 7 DISCUSSION AND FUTURE WORK

We have demonstrated that simple game rules, multi-agent competition, and standard reinforcement learning algorithms at scale can induce agents to learn complex strategies and skills. We observed emergence of as many as six distinct rounds of strategy and counter-strategy, suggesting that multi-agent self-play with simple game rules in sufficiently complex environments could lead to open-ended growth in complexity. We then proposed to use transfer as a method to evaluate learning progress in open-ended environments and introduced a suite of targeted intelligence tests with which to compare agents in our domain.

Our results with hide-and-seek should be viewed as a proof of concept showing that multi-agent autocurricula can lead to physically grounded and human-relevant behavior. We acknowledge that the strategy space in this environment is inherently bounded and likely will not surpass the six modes presented as is; however, because it is built in a high-fidelity physics simulator it is physically grounded and very extensible. In order to support further research in multi-agent autocurricula, we are open-sourcing our environment code.

Hide-and-seek agents require an enormous amount of experience to progress through the six stages of emergence, likely because the reward functions are not directly aligned with the resulting behavior. While we have found that standard reinforcement learning algorithms are sufficient, reducing sample complexity in these systems will be an important line of future research. Better policy learning algorithms or policy architectures are orthogonal to our work and could be used to improve sample efficiency and performance on transfer evaluation metrics.

We also found that agents were very skilled at exploiting small inaccuracies in the design of the environment, such as seekers surfing on boxes without touching the ground, hiders running away from the environment while shielding themselves with boxes, or agents exploiting inaccuracies of the physics simulations to their advantage. Investigating methods to generate environments without these unwanted behaviors is another import direction of future research (Amodei et al., 2016; Lehman et al., 2018).

### ACKNOWLEDGMENTS

We thank Pieter Abbeel, Rewon Child, Jeff Clune, Harri Edwards, Jessica Hamrick, Joel Liebo, John Schulman and Peter Welinder for their insightful comments on this manuscript. We also thank Alex Ray for writing parts of our open sourced code.

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

# Appendix

## Table of Contents

## A    FURTHER EMERGENCE RESULTS

### A.1    TRAJECTORY TRACES FROM EACH STAGE OF EMERGENT STRATEGY

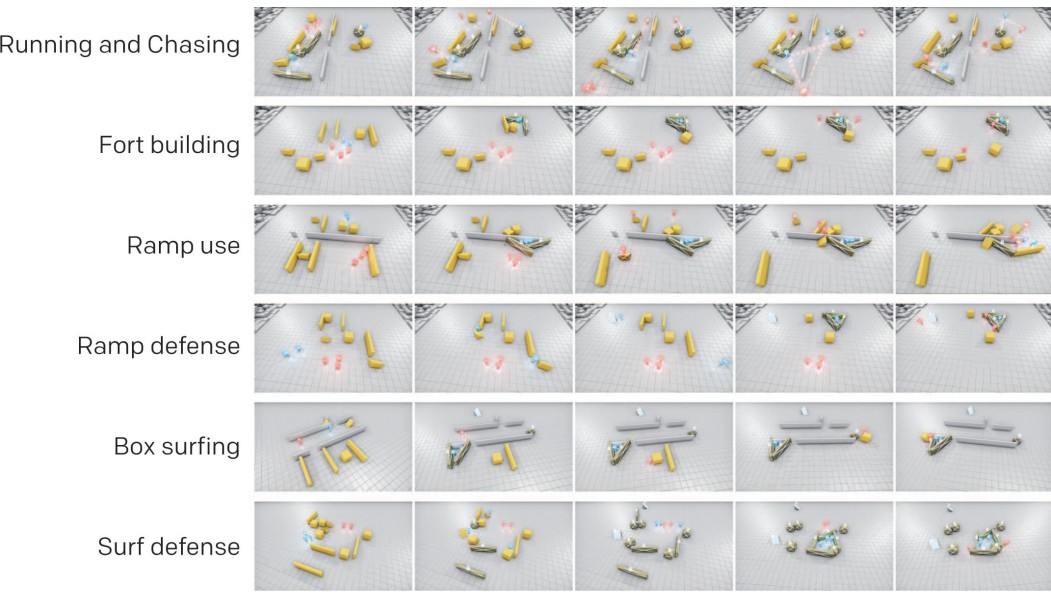

Figure A.1: Trajectory Traces From Each Stage of Emergent Strategy. Rows correspond to different strategies, columns to characteristic snapshots in chronological order within an episode that showcase the strategy. *Running and chasing*: Hiders (blue) try to hide behind walls and moveable objects, and seekers (red) chase and search for hiders. *Fort building:* During the preparation phase (first 3 snapshots), hiders build a fort out of 3 boxes and hide inside. Seekers are not able to overcome this barrier during the remainder of the trial. *Ramp use:* During the preparation phase, hiders build a fort out of 2 boxes and a wall (1st and 2nd snapshot). After the preparation phase, a seeker grabs a ramp (3rd snapshot), drags it towards the fort (4th snapshot) and uses it to enter the fort (5th snapshot). *Ramp defense:* Hiders prevent ramp use by dragging ramps away and locking them in place during the preparation phase (1st, 2nd and 3rd snapshot). They then build a fort in safe distance from the ramps (4th snapshot). After the preparation phase, seekers are not able to enter the fort (5th snapshot). *Box surfing:* A single hider locks both ramps in place and hides inside a fort built from 2 boxes and an interior wall during the preparation phase (1st and 2nd snapshot). The seeker grabs a square box (3rd snapshot) and drags it next to one of the locked ramps to jump on top (4th snapshot). It then "surfs" on the box to the edge of the fort and jumps inside (5th snapshot). *Surf defense*: During the preparation phase, hiders lock all ramps and boxes in place (1st, 2nd and 3rd snapshot) and construct a fort out of 3 boxes (4th snapshot). After the preparation phase, seekers are not able to enter the fort (5th snapshot).

### A.2    DEPENDENCE OF SKILL EMERGENCE ON RANDOMNESS IN THE TRAINING DISTRIBUTION OF ENVIRONMENTS

Discovering tool use is a difficult exploration problem, as only a very specific set of actions will lead to an increase in reward. During training, we find that randomization over many components of our environment, such as the number of agents in each team, the number of boxes, the shape of boxes (square or elongated), the initial location of agents and objects and the presence and location of interior and exterior walls helps emergence, despite leading to a broader training distribution. As we successively reduce the amount of randomization, we find that fewer stages of skill progression emerges, and with at times less sophisticated strategies (e.g. hiders learn to run away and use boxes as moveable shields), Table A.1.

| team size | # boxes | box shape | initial location | walls | emergence |
|-----------|---------|-----------|------------------|-------|-----------|
| 1-3 | 3-9 | random | random | random | 6 stages |
| 1 | 3-9 | random | random | random | 6 stages |
| 1 | 7 | fixed | random | random | 6 stages |
| 1-3 | 3-9 | random | random | no walls | 4 stages |
| 1 | 3-9 | random | random | no walls | 2 stages* |
| 1 | 7 | fixed | random | no walls | 2 stages* |
| 1 | 7 | fixed | fixed | no walls | 2 stages |

* Hiders run away and use boxes as movable shields.

Table A.1: Number of stages of emergence for different levels of environmental randomization (batch size is held constant).

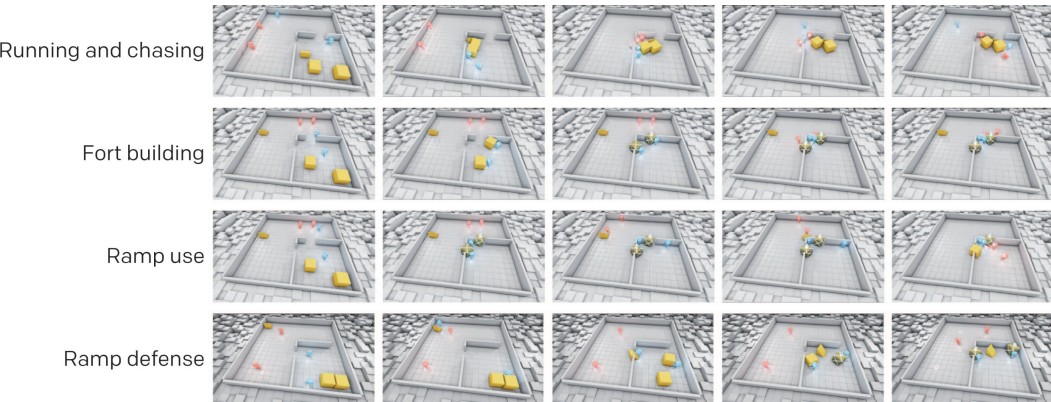

Figure A.2: Sample Trajectory Traces From Each Stage of Emergent Strategy in the Quadrant Environment. Rows correspond to different strategies, columns to characteristic snapshots in chronological order within an episode that showcase the strategy. *Running and chasing*: Hiders (blue) try to hide behind walls and moveable objects, seekers (red) chase and search for hiders. *Fort building:* During the preparation phase (first 3 snapshots), hiders push the two boxes into the two doors of the lower right room and lock them in place. Seekers are not able to overcome this barrier during the remainder of the trial. *Ramp use:* During the preparation phase, hiders secure the room using the two boxes (1st and 2nd snapshot). After the preparation phase, a seeker grabs the ramp (3rd snapshot), drags it towards the walls of the room (4th snapshot) and both seekers use the ramp to enter the room (5th snapshot). *Ramp defense:* Hiders prevent ramp use by grabbing the ramp during the preparation phase (2nd snapshot), pulling it inside the room (3rd snapshot), and then secure the doors using the 2 blocks (4th snapshot). After the preparation phase, the seekers are not able to enter the room (5th snapshot). Also, note that while one hider brings the ramp into the room, the other hider drags a box out of corner and then passes it to its team mate (3rd and 4th snapshot), evidence of sophisticated coordination between hiders.

## A.3 QUADRANT ENVIRONMENT

As well as the default *random rooms* environment described above, we also train and evaluate agents in the much simpler *quadrant* environment (Figure A.2) for illustration purposes. This environment consists of a room in the lower right corner that is connected by 1-2 doors to the remainder of the environment. Unlike in the main environment, the interior walls are fixed and only the location of the doors vary from episode to episode. Hiders are spawned randomly in the environment, but seekers are only spawned outside the room. Two cubic boxes are spawned inside the room and one ramp is spawned inside or outside the room. Both hiders and seekers receive a negative reward if they leave the environment or if they bring an object outside the environment. Also, in contrast to the main environment, only boxes, but not ramps can be locked in place.

The quadrant environment gives rise to a similar skill progression as in the main environment (Figure A.2): basic running and chasing, then fort building by hiders, ramp use by seekers, and ramp defense by hiders. Hiders create shelter by pushing boxes into the doors of the room in the lower right corner, preventing seekers from entering. Seekers drag ramps to the exterior walls of the room and then use them to jump inside. As ramps are not lockable in this environment, hiders instead defend against ramp use by pulling the ramp inside the room before securing the wall opening with boxes.

## A.4 FURTHER ABLATIONS

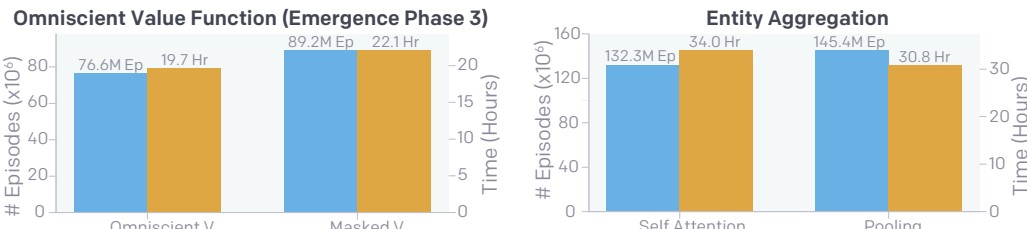

Figure A.3: Effect of Omniscient Value Function and Pooling Architectures on Emergent Autocurricula. In blue we show the number of episodes required and in orange the wall clock time required to achieve stage 4 (ramp defense) of the emergent skill progression presented in Figure 1 for varying batch and model sizes. Note that in our ablation comparing omniscient and masked value functions, the experiment with a masked value function never reached stage 4 in the allotted time, so here we compare timing to stage 3.

In Figure A.3 we compare the performance between a masked and omniscient value function as well a purely pooling architecture versus self-attention. We find that using an omniscient value function, meaning that the value function has access to the full state of the unobscured environment, is critical to progressing through the emergent autocurricula at the given scale. We found that with the same compute budget, training with a masked value function never progressed past stage 3 (ramp usage). We further found that our self-attention architecture increases sample efficiency as compared to an architecture that only embeds and then pools entities together with a similar number of parameters. However, because self-attention requires more compute despite having the same number of parameters, the wall-clock time to convergence is slightly slower.

## A.5 EVALUATING AGENTS AT DIFFERENT PHASES OF EMERGENCE

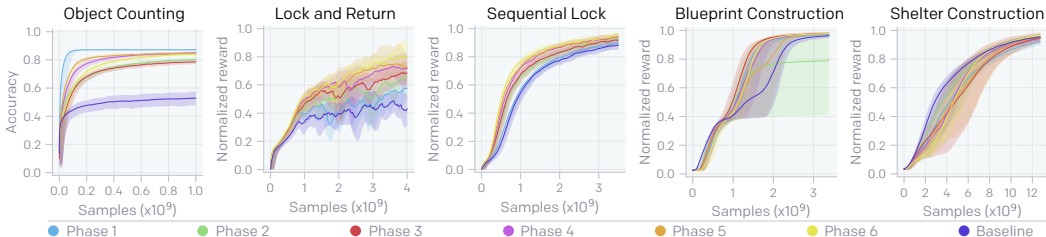

Figure A.4: Fine-tuning From Different Phases in the Emergent Autocurriculum. We plot the mean performance on the transfer suite and 90% confidence interval across 3 seeds and smooth performance and confidence intervals with an exponential moving average. We show the fine-tuning performance for a policy sampled at each of the six phases of emergent strategy (see Figure 1).

In Figure A.4 we evaluate policies sampled during each phase of emergent strategy on the suite of targeted intelligence tasks, by which we can gain intuition as to whether the capabilities we measure improve with training, are transient and accentuated during specific phases, or generally uncorrelated to progressing through the autocurriculum. We find that the hide-and-seek agent improves on the navigation and memory tasks as it progresses; notably on Lock and Return, the performance monotonically increases with emergence phase, and the policy from the phase 6 performs 20% better than the policy from phase 1. However, performance on Object Counting is transient; during

Figure A.5: Example trajectory in hide-and-seek environment with additional food reward. During the preparation phase, hiders lock the ramps at the boundary of the play area (2nd snapshot) and construct a fort around the food reward (3rd and 4th snapshot). After the preparation phase, hiders can eat food and receive the additional food reward, because they are hidden from seekers inside the fort.

phase 1 the hide-and-seek agent performs extremely well, much better than all baselines and other phases but loses this ability in later stages. Finally, we find that performance on the manipulation tasks is relatively uncorrelated to the phases of emergence, and surprisingly the policy transferred from phase 1, the phase before any tool use emerges, performs comparably well to other phases.

## A.6 Alternative Games to Hide-and-Seek with Secondary Objectives

### A.6.1 Hide-and-Seek with Food Reward

In the main hide-and-seek environment, hiders build forts at locations that would give them the best chance of staying hidden from seekers. To test whether hiders could be incentivized by a secondary objective to adapt the location of forts, we add additional food rewards and test whether hiders would bias the location of their forts towards the location of the food. In this environment, hiders can eat food and receive food rewards only under the following conditions: after the preparation phase, when all hiders are hidden from seekers and when the food is sufficiently close and visible to them. Therefore, just chasing after food would not be an effective strategy for hiders, because hiders receive neither hide-and-seek nor food rewards if one or more hiders are seen by a seeker. Instead, hiders are incentivized to build forts around the location of food and then eat the food while being unobserved by seekers (figure A.5).

Food is distributed in the form of 5 food pellets that are spawned close to each other in a rectangular area in the center of the environment whose side length is 1/4 of the room size. Each food pellet can provide a positive food reward of +1 for each time step. Food rewards are shared between hiders, irrespective of which hider eats a food item. Hiders (as well as seekers) can observe the locations of food pellets as separate entities; if food pellets are obstructed or outside the field of view they are masked out like other type of objects.

As shown in figure A.6, this environment gives rise to four levels of skill progression, similar to the one of the main environment: basic running and chasing, then fort building by hiders, ramp use by seekers, and ramp defense by hiders. Moreover, hiders consume food, and food consumption is highly correlated with their ability to construct stable forts; food consumption decreases during the initial phase as seekers get better at chasing and therefore prevent hiders from eating food. Food consumption then increases again as hiders learn to construct forts and shield themselves from the view of seekers, but plateaus once seekers learn to use ramps. Finally, food consumption rises again as hiders get better at defending against ramp use.

### A.6.2 Hide-and-Seek with Dynamic Food

In the food variant introduce in Sec. A.6.1, the location of food pellets is fixed throughout the episode. Here we consider a dynamic food variant such that a food pellet will be eaten up, i.e., disappear, when a hider is close to it, and then a new food pellet will show up in a different location but still within the center region. More precisely, the game area is simply an empty room without outside walls containing 2 seekers, 3 hiders, 1 dynamic food and 8 elongated boxes. The food will be always located within a square in the center of the environment with side length 1/5 of the game area size. We inherit the same reward and policy structures from the previous game.

In this game, merely building a center fort is not sufficient for obtaining the highest reward since the food might disappear and respawn outside the fort. The agent must ensure that the fort is large

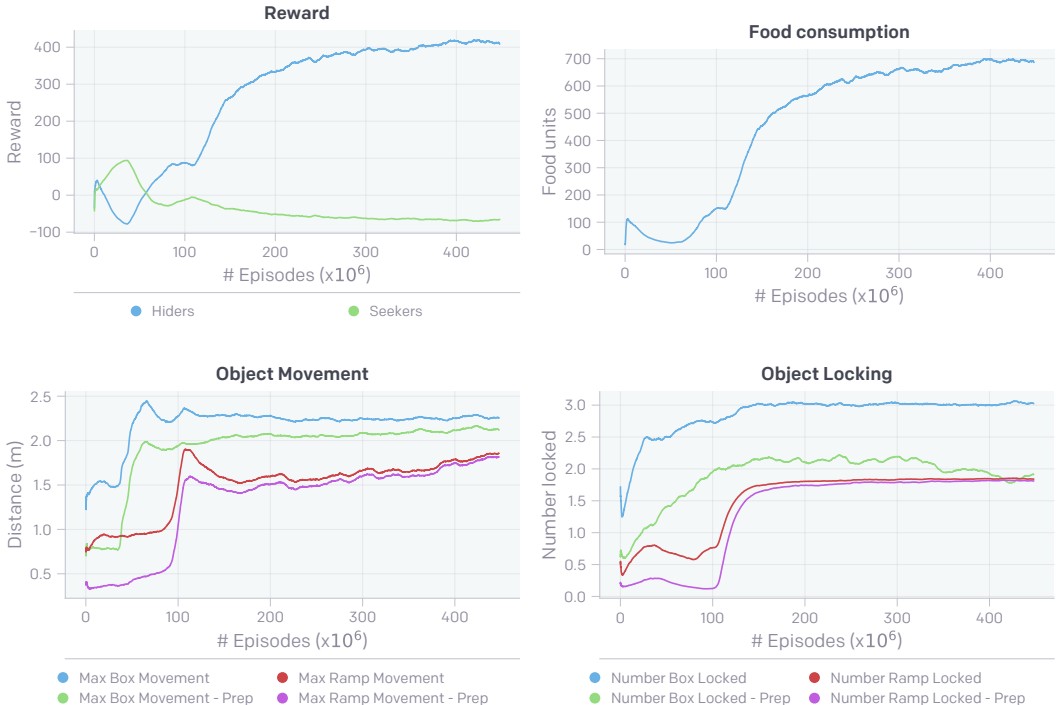

Figure A.6: Reward and environment specific statistics during emergence in hide-and-seek with secondary food rewards for hiders.

enough such that all the possible spawning positions of the food will be inside the fort. Such a behavior does emerge after training for around $4.5 \times 10^{10}$ samples.

We also experimented on variants where the food spawning region has different side length. When the side length is reduced to 1/6 of the game area, the same behavior emerges faster taking $1.5 \times 10^{10}$ samples. However, when the side length of the dynamic food region is increased to 1/4 of the game area, hiders converge to a policy that ignores the food and only builds a small fort to protect themselves.

### A.6.3 FOOD PROTECTION GAME

In the previous two variants, we introduce extra food reward to the hider in addition to the original hide-and-seek reward for promoting more goal-oriented behavior. Now we consider a different game rule such that the competition between hiders and seekers only depends on the food collecting reward and show that this rule can also lead to tool use and complex behavior.

We consider an empty game area surrounded by walls that contains 50 food pellets randomly distributed in a center square of size 2/3 of the game environment. There are 3 hiders, 2 seekers and 7 elongated boxes in the game. The only goal for seekers is to collect food. Once a food pellet is collected by any of the seeker, a +3 reward will be given to all the seekers and then the food will disappear permanently from the game. The goal for the hider is to protect the food from seekers and their reward is simply the negative value of seekers. Each episode consists of 200 time steps. The preparation period, in which only hiders can move, extends to the first 100 time steps. The last 60 steps of the game corresponds to a food collecting period, during which the hiders cannot move. Additionally, after the preparation phase we also add a -1 "boundary penalty" for hiders when they are too close to the wall to ensure they stay within the food region.

Our initial motivation for this task was to promote hiders to learn to construct complex and large fort structures with a more direct competition pressure. For example, we expected that the agents would learn to build a large fort that would cover as much food as possible. Unfortunately, such behavior did not emerge. However, after around $7 \times 10^{10}$ samples of training, the hiders surprisingly discover

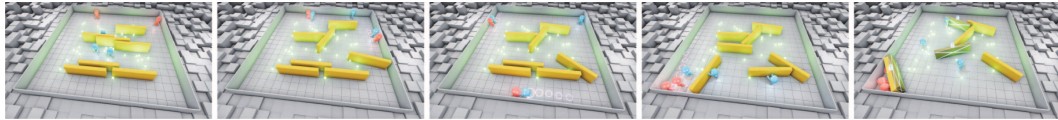

Figure A.7: Example trajectory in the food protection game. During the preparation phase, the hiders first explore the room to find seekers, then physically push the seekers to the corner and eventually collaboratively grab elongated boxes to build a small fort surrounding the seekers to trap them there. To secure the trap, they even learn to build the fort using *two* elongated boxes to prevent seekers from escaping.

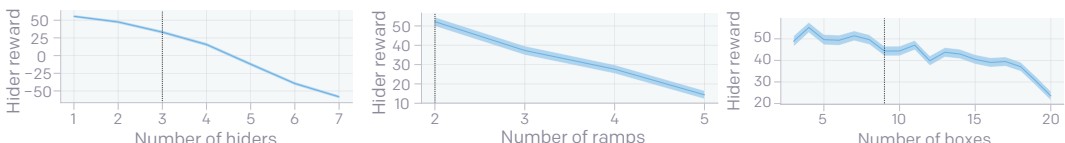

Figure A.8: Zero-shot generalization to a larger number of hiders (left), ramps (center) and boxes (right). The dotted line denotes the boundary of the training distribution (1-3 hiders, 2 ramps, 1-9 boxes). Error bars denote standard error of the mean.

an extremely efficient and effective strategy: they learn to physically 'push' the immobile seekers to the corner of the environment during the preparation phase and then collaboratively use elongated boxes to trap the seekers in the corner. Figure A.7 illustrates a trajectory corresponding to this strategy. Interestingly, the hiders demonstrate strong collaborations. Since there are 2 seekers and 3 hiders, when the 2 hiders are pushing the seekers to the corner, the 3rd hider will follow the 'pushing' hiders distantly, and once the pushing hiders finish the job, this 3rd hider will quickly grab a box to build the trap without blocking the escaping route of those 2 pushing hiders. We emphasize that this behavior was completely surprising to us and further indicates that complex behavior can emerge from multi-agent self-play with a wide range of competition rules.

Lastly, we also experimented with different versions of the close-to-wall "boundary penalties". In all cases, we observe similar fort-building behaviors, namely blocking the seekers to the corner, with tiny variations. When we remove the penalty, the behavior emerges twice faster ($3.5 \times 10^{10}$ samples) and 1 hider will stay in the fort with the seekers instead of getting outside. When we add the penalty throughout the episode, the hiders even learned to stay in the center region while grabbing an elongated box as a tool to push the seekers towards the corner. However, the behavior emerges twice slower in this setting, i.e., $1.4 \times 10^{11}$ samples required.

## A.7 Zero-shot generalization

We find that the trained policies can zero-shot generalize to environments with slightly larger number of hiders, ramps or boxes in Figure A.8. The hider reward provides a reasonably good measure for generalization performance, since the modifications in question make constructing safe shelter more difficult. First, we looked at increasing the number of hiders from 1-3 hiders, as in the training distribution, to 4-6 hiders; increasing the number of hiders decreases the hider reward because all hiders need to stay hidden from seekers. However, the decline in hider reward is very gradual, indicating that the policy generalizes well to more than 4 hiders. A similar effect occurs when increasing the number of ramps because hiders need to secure more ramps from seekers. If we increase the number of ramps from 2 to 3 or 4 the hider reward drops only gradually. Finally, we find hider performance is remarkably stable, though still slowly declines, when increasing the number of boxes.

# B OPTIMIZATION DETAILS

## B.1 NOTATION

We consider the standard multi-agent reinforcement learning formalism of $N$ agents interacting with each other in an environment. This interaction is defined by a set of states $\mathcal{S}$ describing the state of the world and configurations of all agents, a set of observations $\mathcal{O}^1, \ldots \mathcal{O}^N$ of all agents, a set of actions $\mathcal{A}^1, \ldots, \mathcal{A}^N$ of all agents, a transition function $\mathcal{T} : \mathcal{S} \times \mathcal{A}^1 \ldots \mathcal{A}^N \to \mathcal{S}$ determining the distribution over next states, and a reward for each agent $i$ which is a function of the state and the agent's action. Agents choose their actions according to a stochastic policy $\pi_{\theta_i} : \mathcal{O}^i \times \mathcal{A}^i \to [0, 1]$, where $\theta_i$ are the parameters of the policy. In our formulation, policies are shared between agents, $\pi_{\theta_i} = \pi_\theta$ and the set of observations $\mathcal{O}$ contains information for which role (e.g. hider or seeker) the agent will be rewarded. Each agent $i$ aims to maximize its total expected discounted return $R^i = \sum_{t=0}^H \gamma^t r_t^i$, where $H$ is the horizon length and $\gamma$ is a time discounting factor that biases agents towards preferring short term rewards to long term rewards. The *action-value function* is defined as $Q^{\pi_i}(s_t, a_t^i) = \mathbb{E}[R_t^i | s_t, a_t^i]$, while the *state-value function* is defined as $V^{\pi_i}(s_t) = \mathbb{E}[R_t | s_t]$. The *advantage function* $A^{\pi_i}(s_t, a_t^i) := Q^{\pi_i}(s_t, a_t^i) - V^{\pi_i}(s_t)$ describes whether taking action $a_t^i$ is better or worse for agent $i$ when in state $s_t$ than the average action of policy $\pi_i$.

## B.2 PROXIMAL POLICY OPTIMIZATION (PPO)

Policy gradient methods aim to estimate the gradient of the policy parameters with respect to the discounted sum of rewards, which is often non-differentiable. A typical estimator of the policy gradient is $g := \mathbb{E}[\hat{A}_t \nabla_\theta \log \pi_\theta]$, where $\hat{A}$ is an estimate of the advantage function. PPO (Schulman et al., 2017), a policy gradient variant, penalizes large changes to the policy to prevent training instabilities. PPO optimizes the objective $L = \mathbb{E} \left[ \min(l_t(\theta)\hat{A}_t, \text{clip}(l_t(\theta), 1 - \epsilon, 1 + \epsilon)\hat{A}_t \right]$, where $l_t(\theta) = \frac{\pi_\theta(a_t|s_t)}{\pi_{old}(a_t|s_t)}$ denotes the likelihood ratio between new and old policies and $\text{clip}(l_t(\theta), 1 - \epsilon, 1 + \epsilon)$ clips $l_t(\theta)$ in the interval $[1 - \epsilon, 1 + \epsilon]$.

## B.3 GENERALIZED ADVANTAGE ESTIMATION

We use Generalized Advantage Estimation (Schulman et al., 2015) with horizon length $H$ to estimate the advantage function. This estimator is given by:

$$\hat{A}_t^H = \sum_{l=0}^H (\gamma\lambda)^l \delta_{t+l}, \qquad \delta_{t+l} := r_{t+l} + \gamma V(s_{t+l+1}) - V(s_{t+l})$$

where $\delta_{t+l}$ is the TD residual, $\gamma, \lambda \in [0, 1]$ are discount factors that control the bias-variance tradeoff of the estimator, $V(s_t)$ is the value function predicted by the value function network and we set $V(s_t) = 0$ if $s_t$ is the last step of an episode. This estimator obeys the reverse recurrence relation $\hat{A}_t^H = \delta_t + \gamma\lambda \hat{A}_{t+1}^{H-1}$.

We calculate advantage targets by concatenating episodes from policy rollouts and truncating them to windows of $T = 160$ time steps (episodes contain 240 time steps). If a window $(s_0, \ldots, s_{T-1})$ was generated in a single episode we use the advantage targets $(\hat{A}_0^{H=T}, \hat{A}_1^{H=T-1}, \ldots, \hat{A}_{T-1}^{H=1})$. If a new episode starts at time step $j$ we use the advantage targets $(\hat{A}_0^{H=j}, \hat{A}_1^{H=j-1}, \ldots, \hat{A}_{j-1}^{H=1}, \hat{A}_j^{H=T-j}, \ldots, \hat{A}_{T-1}^{H=1})$.

Similarly we use as targets for the value function $(\hat{G}_0^{H=T}, \hat{G}_1^{H=T-1}, \ldots, \hat{G}_{T-1}^{H=1})$ for a window generated by a single episode and $(\hat{G}_0^{H=j}, \hat{G}_1^{H=j-1}, \ldots, \hat{G}_{j-1}^{H=1}, \hat{G}_j^{H=T-j}, \ldots, \hat{G}_{T-1}^{H=1})$ if a new episode starts at time step $j$, where the return estimator is given by $\hat{G}_t^H := \hat{A}_t^H + V(s_t)$. This value function estimator corresponds to the TD($\lambda$) estimator (Sutton & Barto, 2018).

### B.4 NORMALIZATION OF OBSERVATIONS, ADVANTAGE TARGETS AND VALUE FUNCTION TARGETS

We normalize observations, advantage targets and value function targets. Advantage targets are z-scored over each buffer before each optimization step. Observations and value function targets are z-scored using a mean and variance estimator that is obtained from a running estimator with decay parameter $1 - 10^{-5}$ per optimization substep.

### B.5 OPTIMIZATION SETUP

Training is performed using the distributed *rapid* framework (OpenAI, 2018). Using current policy and value function parameters, CPU machines roll out the policy in the environment, collect rewards, and compute advantage and value function targets. Rollouts are cut into windows of 160 timesteps and reformatted into 16 chunks of 10 timesteps (the BPTT truncation length). The rollouts are then collected in a training buffer of 320,000 chunks. Each optimization step consists of 60 SGD substeps using Adam with mini-batch size 64,000. One rollout chunk is used for at most 4 optimization steps. This ensures that the training buffer stays sufficiently on-policy.

### B.6 OPTIMIZATION HYPERPARAMETERS

Our optimization hyperparameter settings are as follows:

| | |
|---|---|
| Buffer size | 320,000 |
| Mini-batch size | 64,000 chunks of 10 timesteps |
| Learning rate | $3 \cdot 10^{-4}$ |
| PPO clipping parameter $\epsilon$ | 0.2 |
| Gradient clipping | 5 |
| Entropy coefficient | 0.01 |
| $\gamma$ | 0.998 |
| $\lambda$ | 0.95 |
| Max GAE horizon length $T$ | 160 |
| BPTT truncation length | 10 |

### B.7 POLICY ARCHITECTURE DETAILS

Lidar observations are first passed through a circular 1D-convolution and concatenated onto the agents representation of self, $x_{\text{self}}$. Each object is concatenated with $x_{\text{self}}$ and then embedded with a dense layer where parameters are shared between objects of the same type, e.g. all boxes share the same embedding weights. All the embedded entities are then passed through a residual self-attention block, similar to Vaswani et al. (2017) but without position embeddings, in the form of $y = \text{dense}(\text{self\_attention}(x)) + x$. We then average-pool entity embeddings and concatenate this pooled representation to $x_{\text{self}}$. Note that in the policy network the entities not observed by each agent are masked away through self-attention and pooling. Finally, this pooled representation is passed through another dense layer and an LSTM (Hochreiter & Schmidhuber, 1997) before pulling off separate action heads for each of the 3 action types described in Section 3. We also add layer normalization (Ba et al., 2016) to every hidden layer of the policy network except the 1D-convolution layer. We empirically observe that layer normalization leads to faster training and better transfer performance.

| | |
|---|---|
| Size of embedding layer | 128 |
| Size of MLP layer | 256 |
| Size of LSTM layer | 256 |
| Residual attention layer | 4 attention heads of size 32 |
| Weight decay coefficient | $10^{-6}$ |

## C  INTELLIGENCE TEST SUITE DETAILS

All evaluation tasks in the intelligence test suite are single-agent tasks. In order to align the observations between the hide-and-seek environment and the evaluation tasks, we add fake hider observations, preparation phase observations and set them all to 0.

### C.1  COGNITION AND MEMORY TASK

All the variations of cognition and memory tasks have a horizon of 120 timesteps, and all boxes are cuboids.

**Object Counting:**

An agent is pinned in place and watches as 6 boxes move either to left or right at random. The boxes disappear behind walls such that agent can no longer see them, and the agent is asked to predict how many boxes went left or right far after all the boxes have disappeared. In this test we evaluate the quality of the *existing* representation by holding the agent's policy fixed and only train a new classification head from the agent's LSTM state. The classification head first passes the LSTM state through a layernorm and a single dense layer with 64 units. We then do a 7-class classification predicting whether 0 through 6 boxes have gone to the left.

**Lock and Return:**

In this game, the agent needs to navigate towards a hidden box, lock it, and then return to its starting position.

The game area has 6 randomly generated connected rooms with static walls and 1 box. When the box is locked, the agent will be given a reward of +5. If the agent unlocks the box during the episode, a -5 penalty will be given. Additionally, if the box remains unlocked at the end of the episode, the agent will be given another -5 penalty. A success is determined when the agent returns to its starting location within 0.1 radius and with the box locked. For promoting fast task accomplishment, we give the agent a +1 reward for each timestep of success. We also introduce shaped reward with coefficient of 0.5 for easier learning: at each time step, the shaped reward is the decrement in the distance between the agent location towards the target (either the unlocked box or the starting location).

**Sequential Lock:**

There are 4 boxes and the agent needs to lock all the boxes in an unobserved order sequentially. A box can be locked only if it is locked in the right order.

The game area is randomly partition into three rooms with 2 walls. The 4 boxes are randomly placed in the game area. Each room has a ramp. The agent has to utilize the ramps to navigate between rooms. When a box is successfully locked (according to the order), a +5 bonus is given. If a box is unlocked, -5 penalty will be added. When all the boxes get locked, the agent will receives a +1 per-timestep success bonus. We also use the same shaped distance reward as the *lock and return* task here.

### C.2  MANIPULATION TASK

All variations of the manipulation task have 8 boxes, but no ramps.

**Construction from Blueprint:**

The horizon is at most 240 timesteps, but an episode can end early if the agent successfully finishes the construction. The game area is an empty room. The locations of the construction sites are sampled uniformly at random (we use rejection sampling to ensure that construction sites do not overlap).

For each construction site, agents observe its position and that of its 4 corners. Since there are no construction sites in the hide-and-seek game and the count-based baseline environments, we need to change our policy architecture to integrate the new observations. Each construction site observation is concatenated with $x_{self}$ and then embedded through a new dense layer shared across all sites. This

dense layer is randomly initialized and added to the multi-agent and count-based policies before the start of training. The embedded construction site representations are then concatenated with all other embedded object representations before the residual self-attention block, and the rest of the architecture is the same as the one used in the hide-and-seek game.

The reward at each timestep is equal to a reward scale constant times the mean of the smooth minimum of the distances between each construction site corner and every box corner. Let there be $k$ construction sites and $n$ boxes, and let $d_{ij}$ be the distance between construction site corner $i$ and box corner $j$, and let $d_i$ be the smooth minimum of the distances from construction site corner $i$ to all box corners. The reward at each timestep follows the following formula:

$$
d_i = \left(\sum_{j=1}^{4n} d_{ij} e^{\alpha d_{ij}}\right) / \sum_{j=1}^{4n} e^{\alpha d_{ij}} \quad \forall i = 1, 2, \ldots, 4k
$$

$$
rew = s_d \left(\sum_{i=1}^{4k} d_i\right) / 4k
$$

Here, $s_d$ is the reward scale parameter and $\alpha$ is the smoothness hyperparameter ($\alpha$ must be non-positive; $\alpha = 0$ gives us the mean, and $\alpha \to -\infty$ gives us the regular min function). In addition, when all construction sites have a box placed within a certain distance $d_{min}$ of them, and all construction site corners have a box corner located within $d_{min}$ of them, the episode ends and all agents receive reward equal to $s_c * k$, where $s_c$ is a separate reward scale parameter. For our experiment, $n = 8$ and $k$ is randomly sampled between 1 and 4 (inclusive) every episode. The hyperparameter values we use for the reward are the following:

$$
\alpha = -1.5
$$
$$
s_d = 0.05
$$
$$
d_{min} = 0.1
$$
$$
s_c = 3
$$

**Shelter construction:** The goal of the task is to build a shelter around a cylinder that is randomly placed in the play area. The horizon is 150 timesteps, and the game area is an empty room. The location of the cylinder is uniformly sampled at random a minimum distance away from the edges of the room (this is because if the cylinder is too close to the external walls of the room, the agents are physically unable to complete the whole shelter). The diameter of the cylinder is uniformly randomly sampled between $d_{min}$ and $d_{max}$. There are 3 movable elongated boxes and 5 movable square boxes. There are 100 rays that originate from evenly spaced locations on the bounding walls of the room and target the cylinder placed within the room. The reward at each timestep is $(-n * s)$, where $n$ is the number of raycasts that collide with the cylinder that timestep and $s$ is the reward scale hyperparameter.

We use the following hyperparameters:

$$
s = 0.001
$$
$$
d_{min} = 1.5
$$
$$
d_{max} = 2
$$

# D    INTRINSIC MOTIVATION METHODS

We inherit the same policy architecture as well as optimization hyperparameters as used in the hide-and-seek game.

Note that only the *Sequential Lock* task in the transfer suite contains ramps, so for the other 4 tasks we remove ramps in the environment for training intrinsic motivation agents.

## D.1    COUNTED-BASED EXPLORATION

For each real value from the continuous state of interest, we discretize it into 30 bins. Then we randomly project each of these discretized integers into a discrete embedding of dimension 16 with

integer value ranging from 0 to 9. Here we use discrete embeddings for the purpose of accurate hashing. For each input entity, we concatenate all its obtained discrete embeddings as this entity's feature embedding. An max-pooling is performed over the feature embeddings of all the entities belonging to each object type (i.e., agent, lidar, box and ramp) to obtain a entity-invariant object representation. Finally, concatenating all the derived object representations results in the final state representation to count.

We run a decentralized version of the count-based exploration where each parallel rollout worker shares the same random projection for computing embeddings but maintains its own counts. Let $N(S)$ denote the counts for state $S$ in a particular rollout worker. Then the intrinsic reward is calculated by $\frac{0.1}{\sqrt{N(S)}}$.

## D.2 RANDOM NETWORK DISTILLATION

Random Network Distillation (RND) (Burda et al., 2019b) uses a fixed random network, i.e., a target network, to produce a random projection for each state while learns anther network, i.e., a predictor network, to fit the output from the target network on visited states. The prediction error between two networks is used as the intrinsic motivation.

For the random target network, we use the same architecture as the value network except that we remove the LSTM layer and project the final layer to a 64 dimensional vector instead of a single value. The architecture is the same for predictor network. We use the squared difference between predictor and target network output with an coefficient of 1.0 as the intrinsic reward.

## D.3 FINE-TUNING FROM INTRINSIC MOTIVATION VARIANTS

We compare the performances of different policies pretrained by intrinsic motivation variants on our intelligence test suites in Figure D.1. The pretraining methods of consideration include RND and 3 different count-based exploration variants with different state representations. For policies trained by count-based variants, as discussed in the Section 6.1, we know that more concise state representation for counts leads to better emergent object manipulation skills, i.e., only using object position is better than using position and velocity information while using all the input as state representation performs the worst. In our transfer task suites, we observe that polices with better pretrained skills perform better in 3 of the 5 tasks except the *Object Counting* task and the *Construction from Blueprint* task. In *Construction from Blueprint*, policies with better skills adapt better in the early phase but may have a higher chance of failure later in this challenging task. In *Object Counting*, the polices with better skills perform worse. Interestingly, this observation is consistent with the transfer result in the main paper (Figure 6), where the policy pretrained by count-based exploration outperforms the multi-agent pretrained policy. We conjecture that the *Object Counting* task examines some factors of the agents that may not strongly relate to the quality of emergent skills, such as navigation and tool use.

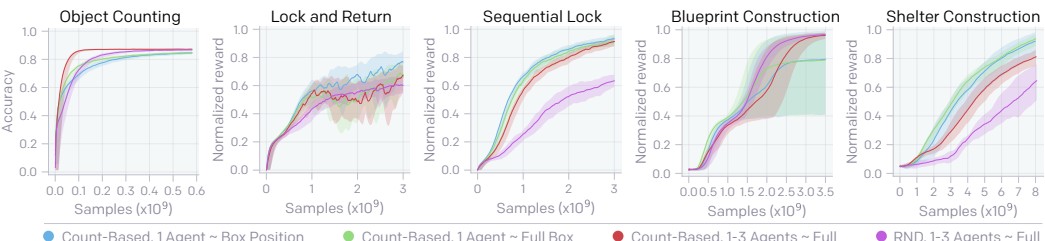

Figure D.1: Fine-tuning From Intrinsic Motivation Variants. We plot the mean performance on the suite of transfer tasks and 90% confidence interval across 3 seeds and smooth performance and confidence intervals with an exponential moving average. We vary the state representation used for collecting counts: in blue we show the performance for a single agent where the state is defined on box 2-D positions, in green we show the performance of a single agent where the state is box position but also box rotation and velocity, in red we show the performance of 1-3 agents with the full observation space given to a hide-and-seek policy, and finally in purple we show the performance also with 1-3 agents and a full observation space but with RND which should scale better than count-based exploration.

