# OpenReview forum: "Emergent Tool Use From Multi-Agent Autocurricula"
_ICLR.cc/2020/Conference — Accept (Spotlight)_

### Official Review · AnonReviewer2 · 2019-10-22
**Official Blind Review #2**

**Rating:** 6

**Review:**

# Review ICLR20, Emergent Tool Use...

This review is for the originally uploaded version of this article. Comments from other reviewers and revisions have deliberately not been taken into account. After publishing this review, this reviewer will participate in the forum discussion and help the authors improve the paper.

I apologize in advance for being reviewer 2.

## Overall

**Summary**

The article introduces a new multi-agent physics environment called "hide-and-seek". The authors trained agents in this environment and studied the emergence of and changes in strategies. The authors also study the performance of these same agents in new "targeted intelligence tests" compared to training from scratch and compared to agents trained with curiosity.

**Overall Opinion**

I think the environment is very appealing and the paper is overall well-structured and demonstrates novel work. Therefore I'd recommend this paper to be accepted. That being said, there are glaring issues with some of the writing that need to be addressed before I think this work conforms to the standards of ICLR. However, if these issues are addressed, I have no issue increasing my review score.

Main problems:

- The majority of the paper presents essentially a case study of what happened during a single seed of policy training. For RL literature that's very uncommon and I think it's consensus that DRL is very sensitive to random seeds. I know that you do have additional seeds in the appendix, but why didn't you mention those in the main body of the paper? You seem to have found some robustness against multiple seeds, so why not show it? And also the fact that Figure 1 & 3 only apply to 1 seed is not mentioned. I think this is easy enough to fix - I suggest since you're already at 10 pages, to just bring in the additional seeds from the appendix and average over their performance in Fig.1&3.
- The contributions section is overselling the work: (1) states that autocurricula lead to changes in agent strategy - Maybe I'm mistaken here but that sounds like a tautology. In other words, "a self-generated sequence of challenges" ("Autocurricula", according to [Leibo et al., 2019][1]) lead to changes in strategy. And (3) advertises "a proposed framework for evaluating agents in open-ended environments" and also "a suite of targeted intelligence tests for our domain". The former of those two is either not in the paper or you mean your section "6.2 Transfer and Fine-Tuning as Evaluation", which isn't novel (see e.g. [Alain & Bengio, 2016][2])
- Your acknowledgments should be anonymized until publication. Otherwise, reviewers might draw conclusions which group published this work, thus violating the double-blind review procedure.

[1]: https://arxiv.org/pdf/1903.00742.pdf
[2]: https://arxiv.org/pdf/1610.01644.pdf

Like I mentioned above, I think these are all easy to address, which should allow acceptance of this work. Here are some additional questions, comments, and nitpicks:

## Specific comments and questions

### Abstract

- "evidence that ... competition may scale better with increasing environment complexity" - that's only shown in the appendix

### Intro

- You mention TD-Gammon as a game, but I think it's an algorithm for the game Backgammon, similarly to how "Go" is the game and "AlphaGo" is an algorithm for playing.

### Rel. Work

- all good

### Hide And Seek

- Arena boundaries: What's the penalty and what's " too far outside the play area"? And in all depictions, it looks like the geometry of the arena is elevated around the edges and the agents don't have a jump action, so how would they ever go out of borders? After watching the videos: Apparently, the jagged-looking arena boundary in the videos is purely cosmetic and agents can still access that space. This is unclear from just the paper and the renderings in Figure 1.

### Policy Optimization

- Policy network and fusion are underspecified: How do you deal with the varying number of agents, boxes, obstacles? Do you just set the x/v of the missing pieces to zero or is the observation actually of a different shape in case there are more/fewer objects or agents? How's the embedding done that is depicted in Figure 2? Also, I didn't see the embedding being mentioned in the text - any reason for that?
- Figure 2 - This diagram is visually appealing but confusing and needs to be improved. Why does the agent's embedding say "1"? Why do the other agents' embeddings have a "-1" at the end of the orange box and the others don't? Is the agent's embedding concatenated with the other embeddings? If so, why and how (concat, sum, multiply, conditional batch norm, etc.)? In the center and on the right you use a blue block to indicate the agent's embedding and then at the bottom right you seem to use it as a network component or something (between the "LSTM")? If you're trying to signal that this is the agent's perception at different stages in the network, I'd use a different color to separate it from the agent's lidar and pos/vel. You don't mention that "x,v" stands for "position, velocity".

### Auto-curriculum and Emergent Behavior

- Figure 3: "environment-specific" (add dash). Draw skill development boundaries like in Fig.1.
- How exactly does the "surfing" work? The seekers step (not jump, right, since there is no jumping?) onto the boxes and then what? The momentum propels the box forward? Do other seekers push the box? Their movement on top of the box somehow moves the box (this seems to be the case judging by the videos but this is the least physically plausible)? This is a super interesting adaptation but I'd suspect the physics simulation to have a bug/glitch that's being exploited here.
- You mention in footnote 3 that the developmental stage and changes in reward aren't necessarily correlated. The same seems to be true for the metrics in Fig.3, which raises the question how did you come up with those boundaries for the different developmental stages in Fig.1? Did someone look at rollouts from the trained policy every couple of million steps? And do all agents learn new skills at the same time or is there a delay? From my understanding, they are all using the same policy and critic networks but maybe dependent on the proximity of an agent to an object/obstacle, it's easier or harder to execute.

### Evaluation

- clear and well-written, slightly too much content in the appendix and not enough in the main paper. Weird appendix numbering - A.6 appears in the main paper pages after A.7

### Discussion and Future Work

- all good

### Appendix

- I appreciate the TOC. I did not look into Appendix B-D because it's another 10 pages on top of the 10 pages of the article.

All in all an interesting work. Good luck with the rebuttal/discussion.

**Experience Assessment:**

I have published one or two papers in this area.

**Review Assessment: Checking Correctness Of Derivations And Theory:**

N/A

**Review Assessment: Checking Correctness Of Experiments:**

I carefully checked the experiments.

**Review Assessment: Thoroughness In Paper Reading:**

I read the paper thoroughly.

---

> ### Author Response · Authors · 2019-11-10
> **Response to Official Review #2 (Part 1)**
>
> Thank you for the very detailed review and constructive criticisms!
>
> — “The majority of the paper presents essentially a case study of what happened during a single seed of policy training...”
>
> Great point, we will update Figures 1 and 3 to be the average across the 3 seeds we show in the appendix. We found very little seed dependence throughout the project, which is likely why we made this oversight.
>
> — “The contributions section is overselling the work: (1) states that autocurricula lead to changes in agent strategy - Maybe I'm mistaken here but that sounds like a tautology. In other words, "a self-generated sequence of challenges" ("Autocurricula", according to [Leibo et al., 2019][1]) lead to changes in strategy.”
>
> In this sentence we were trying to place emphasis on “distinct and compounding phase shifts” — we agree with you that autocurricula by definition are causing changes in strategy, but there is no guarantee that they are distinct shifts (they could just as easily be small changes in strategy). Distinct shifts make it easier to see the effects of an autocurriculum, as small shifts can be hard to detect or analyse. We’ve changed this clause to “clear evidence that multi-agent self-play can lead to emergent autocurricula with many distinct and compounding phase shifts in agent strategy”.
>
> — “And (3) advertises "a proposed framework for evaluating agents in open-ended environments" and also "a suite of targeted intelligence tests for our domain". The former of those two is either not in the paper or you mean your section "6.2 Transfer and Fine-Tuning as Evaluation", which isn't novel (see e.g. [Alain & Bengio, 2016][2])”
>
> After re-reviewing this sentence we agree with you in that it was misleading; we did not intend claim transfer as our idea but rather that we would like to use transfer to evaluate skill progression in open-ended environments. The reason we think it is a contribution is that in most MARL settings, progress is evaluated through play against humans or through metrics like ELO against past versions or other populations. We will modify it to “a proposal to use transfer as a framework for evaluating agents in open-ended environments...”.
>
> —  “Your acknowledgments should be anonymized until publication. Otherwise, reviewers might draw conclusions which group published this work, thus violating the double-blind review procedure.”
>
> Thank you for pointing this out!
>
> —  “"evidence that ... competition may scale better with increasing environment complexity" - that's only shown in the appendix”
>
> We believe Figure 5 is also evidence for this, as you increase the observation space complexity, meaningful interaction with objects goes down when you use intrinsic motivation methods.
>
> — “You mention TD-Gammon as a game, but I think it's an algorithm for the game Backgammon, similarly to how "Go" is the game and "AlphaGo" is an algorithm for playing.”
>
> Thank you for catching this!
>
> —  “Arena boundaries: What's the penalty and what's " too far outside the play area"?”
>
> Great point. We will update the paper with more clear language around this. We give a -10 reward if the agents go outside an 18 meter square (which is 9 times the area of the quadrant game shown in the appendix).
>
> — “Policy network and fusion are underspecified: How do you deal with the varying number of agents, boxes, obstacles? Do you just set the x/v of the missing pieces to zero or is the observation actually of a different shape in case there are more/fewer objects or agents? How's the embedding done that is depicted in Figure 2? Also, I didn't see the embedding being mentioned in the text - any reason for that?”
>
> We have more detail in appendix section B.7, which answers some of your questions, but we will move some of these details to the main text/caption of Figure 2 and add more clarification. The architecture is an attention and pooling based architecture so it naturally deals with varying numbers of objects. We mask out anything not visible to the agent in the attention and pooling operations so that they do not receive privileged information. The embedding weights are shared within object type, e.g. all box entities pass through the same shared embedding function. We currently show in figure 2 that the observations pass through fully connected layers to create these embeddings, but we’ll add the comment about shared weights to the caption.
>
> — “Why does the agent's embedding say "1"? Why do the other agents' embeddings have a "-1" at the end of the orange box and the others don't?”
>
> The agents have an ego-centric architecture, so that “1” shows that that entry is the agent’s observation of itself (the agent itself is just 1 entity as opposed to boxes or other agents which are many entities). There are (# agents - 1) other agents (from the view of any single agent). We’ll add this clarification to the figure caption.

---

> ### Author Response · Authors · 2019-11-10
> **Response to Official Review #2 (Part 2)**
>
> — “Is the agent's embedding concatenated with the other embeddings? If so, why and how (concat, sum, multiply, conditional batch norm, etc.)?”
>
> We concatenate all the entities together, such that the tensor has shape (number entities, entity dimension), run residual self attention, and then average pool getting a fixed sized vector of size (entity dimension). We’ll add this clarification to the text.
>
> — “In the center and on the right you use a blue block to indicate the agent's embedding and then at the bottom right you seem to use it as a network component or something (between the "LSTM")? If you're trying to signal that this is the agent's perception at different stages in the network, I'd use a different color to separate it from the agent's lidar and pos/vel.”
>
> All the colored blocks represent activations in this diagram, but we agree re-using the blue coloring could be confusing. We’ll change the color of the final two blue blocks for the camera ready version of the paper.
>
> — “You don't mention that "x,v" stands for "position, velocity".”
>
> Good catch! Thank you, we’ll add a clarification to the text.
>
> — “Figure 3: "environment-specific" (add dash). Draw skill development boundaries like in Fig.1.”
>
> Good suggestions. Thank you!
>
> — “How exactly does the "surfing" work? The seekers step (not jump, right, since there is no jumping?)”
>
> We do classify this as an exploit of the rules we designed in the last paragraph of Section 7, but we will add more clarification on how it works and that it is an exploit of our intended game rules in Section 5. You are right in that they more “step” or “launch” themselves from a ramp to the box. Once on top of the box, they can still “grab” the box, which keeps the relative orientation and position between agent and box fixed. The agents’ movement action puts a force on the agent regardless of whether the agent is on the ground or not. So if the agent does this while grabbing the box, they will both move together since they have a fixed relative orientation and position.
>
> — “You mention in footnote 3 that the developmental stage and changes in reward aren't necessarily correlated. The same seems to be true for the metrics in Fig.3, which raises the question how did you come up with those boundaries for the different developmental stages in Fig.1? Did someone look at rollouts from the trained policy every couple of million steps?”
>
> We used a combination of looking at the reward, behavioral statistics (Figure 3), and watching trajectories. It is a very interesting line of future research to automatically detect large shifts in agent strategy!
>
> “And do all agents learn new skills at the same time or is there a delay? From my understanding, they are all using the same policy and critic networks but maybe dependent on the proximity of an agent to an object/obstacle, it's easier or harder to execute.”
>
> — All agents have the same weights so they would learn the skill at the same time. However, we’ve run some experiments where each agent has a different policy and we did not notice any significant differences to the shared weight case. Using shared weights is simpler to implement and cheaper to train, which is why we use them for all of our experiments. Anecdotally, agents’ seems to learn the skills for easier cases in the environment first; for instance we notice they often learn to construct a 1 block barricade using existing walls before they learn to construct a 3 block fort in the center of the room.

---

### Official Review · AnonReviewer3 · 2019-10-23
**Official Blind Review #3**

**Rating:** 8

**Review:**

Authors in introduce a new competitive/cooperative physics-based environment in which different teams of agents compete in a visual concealment and search task with visibility-based team-based rewards (although There are no explicit incentives for agents to interact with objects in the environment). They show that, complex behaviour emerge as the episode progresses and agents are able to learn 6 emergent skills/(counter-)strategies (including tool use), where agents intentionally change their environment to suit their needs. Agents trained using self-play

In my opinion, this is an excellent paper which main contribution is to provide experimental evidence that relevant and complex skills and strategies can emerge from multi-agent RL competing scenarios.

Minor comments:

- Hide&seek rules and safety issues: is it not supposed that hiders and the seekers could not get together (i.e., hiders cannot push seekers or as we can see in some videos)? Furthermore, it is surprising (one would say worrying) that hiders identified the barriers as an impediment to the seeker (not only as a way to hide). I wouldn’t say that this is a “ human-relevant strategies and skills “ as the authors claim. Hider agents even double walled seekers!

- Have the authors thought about joining the Animal-AI Olympics (http://animalaiolympics.com/) competition? It would be a great opportunity to to test the skills of your agents in a further general testing scenario. They provide an arena (test-bed) which contains 300 different intelligent tests for testing the cognitive abilities of RL agents (https://www.mdcrosby.com/blog/animalaiprizes1.html) which have to interact with the environment.


**Experience Assessment:**

I have read many papers in this area.

**Review Assessment: Checking Correctness Of Derivations And Theory:**

N/A

**Review Assessment: Checking Correctness Of Experiments:**

N/A

**Review Assessment: Thoroughness In Paper Reading:**

N/A

---

> ### Author Response · Authors · 2019-11-10
> **Response to Official Review #3**
>
> Thank you for the review and questions!
>
> — “Hide&seek rules and safety issues: is it not supposed that hiders and the seekers could not get together (i.e., hiders cannot push seekers or as we can see in some videos)? Furthermore, it is surprising (one would say worrying) that hiders identified the barriers as an impediment to the seeker (not only as a way to hide). I wouldn’t say that this is a “ human-relevant strategies and skills “ as the authors claim. Hider agents even double walled seekers!”
>
> In the environment as is, the hiders can push the seekers during the preparation phase. It’s unclear that this is bad, but we agree that we could easily make it not the case, though it likely would not change the skill progression in the main hide-and-seek environment. However, as you note, this can definitely change the resulting skill progression in other game variants (Figure A.8). We also believe finding methods that can make agents converge on safe outcomes is an important direction for future research!
>
> “Have the authors thought about joining the Animal-AI Olympics (http://animalaiolympics.com/) competition?”
>
> — We thought this would be outside the scope of our current work, but we agree this challenge is very interesting. However, from the description it feels slightly different than the transfer tasks we propose. They say “The goal will always be to retrieve the same food items by interacting with previously seen objects,” where in our transfer tests agents are given very different objectives from the original objective of hide-and-seek.

---

### Official Review · AnonReviewer1 · 2019-10-23
**Official Blind Review #1**

**Rating:** 3

**Review:**

1. Summary

The authors report on an empirical study of emergent behavior of multiple RL agents learning to play hide-and-seek (a sparse reward task). The main point of this paper is that RL agents learning at scale (large number of samples, batch-size 64000). can learn to solve tasks with strategies that are human-interpretable (e.g., using ramps, boxes). Scale also requires various simplifications (e.g., keeping the learning setup as close as possible to a single-agent problem as possible).

Agents are grouped in 2 teams (seekers, hiders). Each agent receives a team reward, e.g., it can be punished for events that it did not participate in, e.g., if a team-mate is seen by an opponent. If hiders are hidden, seekers also automatically see reward. The first 40% of the episode there is no reward to let hiders hide.

There is one actor model, all agents share weights. Hence this is self-play: hiders and seekers use the same agent model. Also, all agents use a central value function that can see the entire state (decentralized execution, centralized learning). This makes the setting basically a single-agent problem, with the only decentralized aspect being each actor model only receiving its own observation. Note that a large body of multi-agent RL work in fact uses agents that do not share weights, etc.

Other features described:
- Auto-curricula: e.g. agents find new strategies (using ramps, boxes) that other agents have to counteract.
- Human-relevant skills: They report that the agent model learns multiple ways to interact with (objects in) the environment that are semantically interesting (resembles something humans might do).
- Authors compare with policies learning via intrinsic motivation.
- Evaluation through transfer learning shows some benefit of transfer of hide-seek agents to auxiliary tasks. However, it is not so clear how this evaluation informs future work on transfer learning (e.g., how would you pick evaluation tasks for a given train-task?)

1. Decision (accept or reject) with one or two key reasons for this choice.

Reject.

The main point of the paper is empirical RL at scale. Although the learned behaviors are human-interpretable, this does not seem surprising given the fact that in many (large-scale) RL applications (Atari games, Go, DotA 2, Starcraft), it has been observed that RL agents can learn to manipulate and use their environment (which includes other agents!) in unexpected ways / find creative ways to exploit the reward function (see e.g. demos in https://www.alexirpan.com/2018/02/14/rl-hard.html). There has also been work on object-level RL [Agnew, Domingos 2018], which involves agents interacting with objects in the environment. Compared to this, the observation that RL agents learn human-interpretable uses of objects does not seem surprising.

The paper also does not give new insights in how to make large-scale RL ``'work'. For instance, there are no significant differences in algorithm / model structure from DotA / Starcraft agents that can inform future large-scale experiments.

The paper also does not introduce new concrete evaluation metrics that can apply to other tasks / RL problems, skill detection / segmentation methods to learn the structure of auto-curricula. Furthermore, the setup is very close to a single-agent problem (see above), and is far simpler in the multi-agent assumptions from other decentralized multi-agent work (Foerster 2018, Jacques 2019, etc).

**Experience Assessment:**

I have published one or two papers in this area.

**Review Assessment: Checking Correctness Of Derivations And Theory:**

N/A

**Review Assessment: Checking Correctness Of Experiments:**

I carefully checked the experiments.

**Review Assessment: Thoroughness In Paper Reading:**

N/A

---

> ### Public Comment · ~Hassam_Sheikh1 · 2019-11-05
> **Main point of the paper is not RL at scale**
>
> Disclaimer: I am neither an author nor in anyway related to OpenAI.
>
> I believe that the main point of this paper is NOT to demonstrate RL at scale, though, as everyone has noticed that work done by OpenAI   mostly requires stupendous amount of compute power (RAPID framework, 128000 cpus) which they have also used here. After reading this paper several times and being a researcher in MARL myself, I believe that judging this paper just on the basis of scale is entirely unfair. The main idea of this paper is the evolution of complex strategies and emergence of auto-curriculum when agents face evolving competition.
>
> The reviewer has mentioned
> "There is one actor model, all agents share weights. Hence this is self-play: hiders and seekers use the same agent model. Also, all agents use a central value function that can see the entire state (decentralized execution, centralized learning). This makes the setting basically a single-agent problem, with the only decentralized aspect being each actor model only receiving its own observation."
> This argument might have carried a lot of weight (pun intended) when the goal of this work is to propose a new SOTA MARL algorithm but does the architecture used here matters? Probably not, I reckon that this architecture probably wont even work for any other standard MARL task.
>
> Secondly "Note that a large body of multi-agent RL work in fact uses agents that do not share weights, etc."  Isn't the citation (Foerster 2018) mentioned at end actually share parameters? The paper mentions " However, we still assume agents have access to opponents’ policy parameters in policy gradient-based LOLA. "
>
> I would not go on explaining what are the intentions of this paper but I can safely say that reviewer has completely missed the point of the paper and just evaluated it on the basis the technical aspects of the paper which are irrelevant for this work.

---

> ### Author Response · Authors · 2019-11-10
> **Response to Official Review #1**
>
> Thank you for your review and constructive criticisms! We’ll try to address each piece of criticism in turn.
>
> — “The main point of the paper is empirical RL at scale”
>
> The main point we hope to convey is that large-scale multi-agent reinforcement learning (MARL) can lead to self-supervised autocurricula in which agents learn successively more complex human-relevant skills such as construction and tool use. We absolutely agree that there have been many amazing previous results from MARL at scale, and we acknowledge many of the works you mention and more in our introduction and related work sections. We believe our work differs from these in that our environment is built from very simple components in a physically grounded simulator, making it extremely extensible. It is much more clear how one could add to or modify the hide-and-seek environment to include more human-relevant components than it is how one could modify games like Go, Dota, or Starcraft.
>
> — “There has also been work on object-level RL … the observation that RL agents learn human-interpretable uses of objects does not seem surprising.”
>
> We agree that there has been much work on object-level RL. We didn’t advertise this as a novel portion of our work, and we’ve already included many citations that use object-level architectures and attention at the end of Section 4. We also acknowledge that there have been prior works where RL learns human-interpretable uses of objects, which is why we include a paragraph in Section 2 on prior work in tool-use; however, our work can be distinguished from these and the work you cite in that we provide no explicit signal for interacting with objects; the pressure to interact with the objects is solely a result of multi-agent competition.
>
> — “The paper also does not give new insights in how to make large-scale RL work”
>
> This paper was not on how to make large-scale RL work, but rather on showing the power of current large-scale RL algorithms in a new setting that is more physically grounded and human-relevant than previous settings like DotA, Starcraft, and Go. The main argument of the paper is that multi-agent autocurricula can lead to agents learning many human-relevant skills like tool-use and construction; the fact that we required no new significant algorithmic modifications actually strengthens this point in our opinion, as the results can’t be confused as a pathology of a new specific algorithm. That being said, we agree that it is a great direction for future research to incorporate methods that can learn faster or better in this environment.
>
> — “The paper also does not introduce new concrete evaluation metrics that can apply to other tasks / RL problems...”
>
> It’s very hard to create transfer tasks that are valid across domains. However, we hope that the tasks we proposed can be used as transfer metrics for any future research within our domain (both of which we will open source).
>
> — “There is one actor model, all agents share weights”
>
> Using shared weights, or at least some portion of training data coming from self-play, is very common (AlphaGo, DotA, Alphastar, Capture-the-Flag, NeuralMMO, etc.), and it doesn’t alter the multi-agent optimization objective. Each agent still takes a greedy gradient and has its own observations and memory state so that at execution they use no privileged information. Shared weights does not mean uni-brain (one brain many actions), which would indeed reduce this to a single agent problem. That being said, we’ve run the hide-and-seek experiment with separate weights for each agent and as expected have seen no difference in learned strategy.
>
> — “all agents use a central value function that can see the entire state. This makes the setting basically a single-agent problem and is far simpler in the multi-agent assumptions from other decentralized multi-agent work”
>
> This is a commonly used method to reduce policy gradient variance in partially observed settings without letting agents cheat at execution time both for MARL (MADDPG, Counterfactual RL, AlphaStar) and also single agent RL (Dactyl, Asymmetric Actor Critic). We ablate this choice in the appendix and find that it is important at the given scale of compute but agents still learn without it.
>
> As for other MARL algorithms, we cite both of the works you mention in our paper already, and it is an excellent line of future research to incorporate methods like these into setups such as hide-and-seek to see if they bring benefit to learning. However, we don’t think algorithmic simplicity is a fault of our work but rather a strength. We show that with only standard simple algorithms, multi-agent autocurricula can lead to human-relevant skills like construction and tool-use in physically grounded environments, which we believe provides a good baseline for future algorithmic research.

---

### Public Comment · ~Murray_Shanahan1 · 2019-10-03
**Great task and fascinating results, but a question about object permanence claims**

Very nice paper. I think the hide-and-seek task is excellent, as it gets at some fundamental common sense concepts (object persistence, obstruction, etc). (It would be especially compelling if the task were solved from pixels.) The sequence of emergent strategies is fascinating. And obviously this is the main result, so the following should be taken in that context.

I have a question re the claim that the object-counting transfer task you propose in Section 6.2 really provides evidence that "the agents have a sense of object permanence". Couldn't the classifier you add on to the pre-trained agent simply count the number of leftward (as opposed to rightward) movements of the boxes? What different does it make, in this task, that the boxes eventually become obscured?

More generally, is it not the case that the agents have an architectural prior that builds in exactly how many objects exist? If I understand Figure 2 correctly, the embedding vector that encodes the objects has a dimension whose length is precisely the number of objects that exist, even when parts of it are masked out. So the permanence of all the objects is, in a sense, built in .

---

> ### Author Response · Authors · 2019-10-11
> **Re: Great task and fascinating results, but a question about object permanence claims**
>
> Thank you for the praise and questions!
>
> According to our understanding, object permanence is typically defined as: the understanding that objects continue to exist even when they cannot be perceived (Piaget J., The construction of reality in the child. Basic Books; New York: 1954.). In the proposed task, the objects are only visible for about 20% of the episode, meaning that for the remaining 80% the agent cannot sense the objects in any way and must make the prediction based only on its memory (in our case the memory unit is an LSTM) of where it saw objects going. The agent must remember how many objects went to one side or the other, so in a sense it is required to understand that objects continue to exist in that area after they have been obscured.  When training on this task, we keep all of the original policy weights, e.g. embedding weights and LSTM weights, fixed such that we are only evaluating the existing representation the agent has after training in hide-and-seek or with intrinsic motivation (the task would be trivial without keeping these weights fixed).
>
> In our policy architecture, all “entities” (meaning objects and other agents)  are pooled together before going into the LSTM. You are correct that before this, there are an equal number of embedding vectors as there are entities in the environment, but this information is lost after masked pooling. For instance, if there are no visible entities at a given timestep, then the output of that masked pooling operation is a vector of 0’s with dimension independent of the number of entities in the game, meaning there is no way for the agent to know how many entities exist past what’s in its memory. We hope this clarifies any confusions on the policy architecture from the current text, and we will try to give more clarification in the next version of the paper.

---

> > ### Public Comment · ~Murray_Shanahan1 · 2019-10-16
> > **Re: Great task and fascinating results, but a question about object permanence claims**
> >
> > Many thanks for the reply

---

### Author Response · Authors · 2019-11-10
**Paper Revisions**

In response to reviews, we’ve made the following updates to the paper:

1. Slightly modified language of contribution statement
2. Changed figures 1 and 3 to plot the mean across 3 seeds and show the seeds
3. Add sentence describing out of bounds condition and reward to Section 3
4. Add more description of policy architecture to caption of Figure 2.
5. Add more description of how/why box surfing is possible to Section 5
6. Remove acknowledgements section (will be added back for camera ready paper)
7. Remove Appendix section on multiple seeds as now figures 1 and 3 include this information
8. Add footnote describing non-shared weights experiments to Section 4.

---

### Decision · Program_Chairs · 2019-12-19

**Decision:**

Accept (Spotlight)

**Comment:**

This paper describes how multi-agent reinforcement learning at scale leads to the evolution of complex behaviors. Actually, "at scale" may be an understatement - a lot of computing power was used here. But the amount of compute used is not the point, rather the point is that complex and fascinating behavior can emerge from a long co-evolutionary process (though gradient-based RL is used here, the principle is the same) where the arms race forms an implicit curriculum. This is the existence proof that people in artificial life and adaptive behavior have been looking for for so long.

Two reviewers were positive about the paper, with a third being negative because the paper does not give any new insights about how to do RL at scale. But that was not the stated aim of the paper, as the authors clarify in a response.

This paper will draw quite some attention and deserves an oral presentation.